# Truncated Affinity Maximization: One-class Homophily Modeling for Graph Anomaly Detection

**Hezhe Qiao, Guansong Pang**[*]
School of Computing and Information Systems, Singapore Management University
hezheqiao.2022@phdcs.smu.edu.sg, gspang@smu.edu.sg

## Abstract

We reveal a *one-class homophily* phenomenon, which is one prevalent property we find empirically in real-world graph anomaly detection (GAD) datasets, *i.e.*, normal nodes tend to have strong connection/affinity with each other, while the homophily in abnormal nodes is significantly weaker than normal nodes. However, this anomaly-discriminative property is ignored by existing GAD methods that are typically built using a conventional anomaly detection objective, such as data reconstruction. In this work, we explore this property to introduce a novel unsupervised anomaly scoring measure for GAD – *local node affinity* – that assigns a larger anomaly score to nodes that are less affiliated with their neighbors, with the affinity defined as similarity on node attributes/representations. We further propose Truncated Affinity Maximization (TAM) that learns tailored node representations for our anomaly measure by maximizing the local affinity of nodes to their neighbors. Optimizing on the original graph structure can be biased by *non-homophily edges* (*i.e.*, edges connecting normal and abnormal nodes). Thus, TAM is instead optimized on truncated graphs where non-homophily edges are removed iteratively to mitigate this bias. The learned representations result in significantly stronger local affinity for normal nodes than abnormal nodes. Extensive empirical results on 10 real-world GAD datasets show that TAM substantially outperforms seven competing models, achieving over 10% increase in AUROC/AUPRC compared to the best contenders on challenging datasets. Our code is available at https://github.com/mala-lab/TAM-master/.

## 1 Introduction

Graph anomaly detection (GAD) aims to identify abnormal nodes that are different from the majority of the nodes in a graph. It has attracted great research interest in recent years due to its broad real-world applications, *e.g.*, detection of abusive reviews or malicious/fraudulent users [9, 16, 32, 37, 57]. Since graph data is non-Euclidean with diverse graph structure and node attributes, it is challenging to effectively model the underlying normal patterns and detect abnormal nodes in different graphs. To address this challenge, graph neural networks (GNNs) have been widely used for GAD. The GNN-based methods are often built using a data reconstruction [8, 11, 31, 59, 65] or self-supervised learning [15, 17, 28, 54, 63] objective. The data reconstruction approaches focus on learning node representations for GAD by minimizing the errors of reconstructing both node attributes and graph structure, while the self-supervised approaches focus on designing a proxy task that is related to anomaly detection, such as prediction of neighbor hops [17] and prediction of the relationship between a node and a subgraph [28], to learn the node representations for anomaly detection.

These approaches, however, ignore one prevalent anomaly-discriminative property we find empirically in real-world GAD datasets, namely *one-class homophily*, *i.e.*, normal nodes tend to have strong connection/affinity with each other, while the homophily in abnormal nodes is significantly weaker

---

[*]Corresponding author: G. Pang

37th Conference on Neural Information Processing Systems (NeurIPS 2023).

than normal nodes. This phenomenon can be observed in datasets with synthetic/real anomalies, as shown in Fig. 1(a) (see App. A for results on more datasets). The abnormal nodes do not exhibit homophily relations to each other mainly because abnormal behaviors are unbounded and can be drawn from different distributions.

Motivated by the one-class homophily property, we introduce a novel unsupervised anomaly scoring measure for GAD – *local node affinity* – that assigns a larger anomaly score to nodes that are less affiliated with their neighbors. Since we do not have access to class labels, we define the affinity in terms of similarity on node attributes/representations to capture the homophily relations within the normal class. Nodes having strong local affinity are the nodes that are connected to many nodes of similar attributes, and those nodes are considered more likely to be normal nodes.

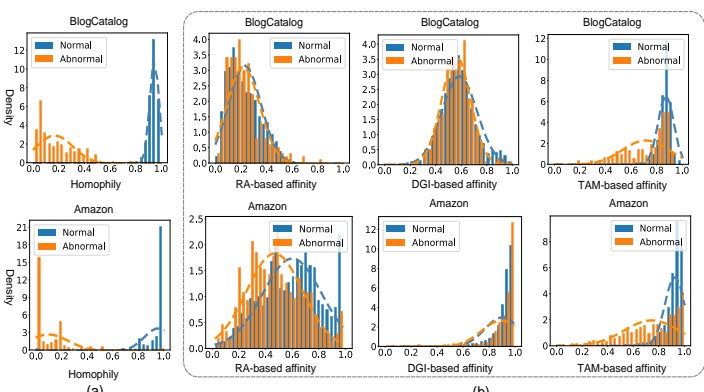

Figure 1: (**a**) Homophily and (**b**) local affinity distributions of normal and abnormal nodes on two popular benchmarks, BlogCatalog [49] and Amazon [10]. The homophily of a given node is calculated using the number of nodes that have the same class label as the given node [13]. The local affinity is calculated on raw attributes (RA) and node representations learned by DGI [51] and TAM, respectively.

One challenge of using this anomaly measure is that some abnormal nodes can also be connected to nodes of similar abnormal behaviors. A straightforward solution to this problem is to apply our anomaly measure in a node representation space learned by off-the-shelf popular representation learning objectives like DGI [51], but the representations of normal and abnormal nodes can become similar due to the presence of *non-homophily edges* (*i.e.*, edges connecting normal and abnormal nodes) that homogenize the normal and abnormal node representations in GNN message passing. To address this issue, we further propose Truncated Affinity Maximization (TAM) that learns tailored node representations for our anomaly scoring measure by maximizing the local node affinity to their neighbors, with the non-homophily edges being truncated to avoid the over-smooth representation issue. The learned representations result in significantly stronger local affinity for normal nodes than abnormal nodes. As shown in Fig. 1(b), it is difficult for using the local node affinity to distinguish normal and abnormal nodes on raw node attributes and DGI-based node representations, whereas the TAM-based node representation space offers well-separable local affinity results between normal and abnormal nodes. In summary, this work makes the following main contributions:

- We, for the first time, empirically reveal the one-class homophily phenomenon that provides an anomaly-discriminative property for GAD. Motivated by this property, we introduce a novel unsupervised anomaly scoring measure, local node affinity (Sec. 3.2).

- We then introduce Truncated Affinity Maximization (TAM) that learns tailored node representations for the proposed anomaly measure. TAM makes full use of the one-class homophily to learn expressive normal representations by maximizing local node affinity on truncated graphs, offering discriminative local affinity scores for accurate GAD.

- We further introduce two novel components (Sec. 3.3), namely Local Affinity Maximization-based graph neural networks (LAMNet for short) and Normal Structure-preserved Graph Truncation (NSGT), to implement TAM. Empirical results on six real-world GAD datasets show that our TAM model substantially outperforms seven competing models.

## 2 Related Work

Numerous anomaly detection methods have been introduced, including both shallow and deep approaches [5, 37], but most of them are focused on non-graph data. There have been many studies on GAD exclusively. Most of previous work use shallow methods [2], such as Radar [22], AMEN [43],

and ANOMALOUS [41]. Matrix decomposition and residual analysis are commonly used in these methods, whose performance is often bottlenecked due to the lack of representation power to capture the rich semantics of the graph data and to handle high-dimensional node attributes and/or sparse graph structures.

GNN-based GAD methods have shown substantially better detection performance in recent years [32]. Although some methods are focused on a supervised setting, such as CARE-GNN [10], PCGNN [27], Fraudre [58], BWGNN [47], and GHRN [12], most of them are unsupervised methods. These unsupervised methods can be generally categorized into two groups: data reconstruction-based and self-supervised-based approach. Below we discuss these most related methods in detail.

**Data Reconstruction-based Approach.** As one of the most popular methods, graph auto-encoder (GAE) is employed by many researchers to learn the distribution of normal samples for GAD. Ding *et al.* [8] propose DOMINANT, which reconstructs the graph structure and node attributes by GAE. The anomaly score is defined as the reconstruction error from both node attributes and its structure. It is often difficult for GAE to learn discriminative representations because it can overfit the given graph data. Several mechanisms, including attention, sampling, edge removing, and meta-edge choosing [9, 27, 30, 45], are designed to alleviate this issue during neighborhood aggregation. Some recent variants like ComGA [31] and AnomalyDAE [11] incorporate an attention mechanism to improve the reconstruction. Other methods like SpaceAE [24] and ResGCN [39] aim to differentiate normal and abnormal nodes that do not have significant deviation, *e.g.*, by exploring residual information between nodes. In general, reconstructing the structure of a node based on the similarities to its neighbors is related to but different from local node affinity in TAM (see Sec. 3.2), and GAE and TAM are also learned by a very different objective (see Sec. 3.3). Being GNN-based approaches, both the reconstruction-based approaches and our approach TAM rely on the node's neighborhood information to obtain the anomaly scores, but we explicitly define an anomaly measure from a new perspective, *i.e.*, local node affinity. This offers a fundamentally different approach for GAD.

**Self-supervised Approach.** Although data reconstruction methods can also be considered as a self-supervised approach, here we focus on non-reconstruction pre-text tasks for GAD. One such popular method is a proxy classification or contrastive learning task [18, 53, 65]. Liu *et al.* [28] propose CoLA, which combines contrastive learning and sub-graph extraction to perform self-supervised GAD. Based on CoLA, SL-GAD is proposed to develop generative attribute regression and multi-view contrastive learning [63]. There are some methods that leverage some auxiliary information such degree, symmetric and hop to design the self-supervised task [6, 17, 23, 42, 56, 61]. For example, Huang *et al.* propose the method HCM-A [17] to utilize hop count prediction for GAD. Although some self-supervised methods construct the classification model on the relationship between the node and the contextual subgraph, this group of methods is not related to local node affinity directly, and its performance heavily depends on how the pre-text task is related to anomaly detection. Similar to the self-supervised approaches, the optimization of TAM also relies on an unsupervised objective. However, the self-supervised approaches require the use of some pre-text tasks like surrogate contrastive learning or classification tasks to learn the feature representations for anomaly detection. By contrast, the optimization of TAM is directly driven by a new, plausible anomaly measure, which enables end-to-end optimization of an explicitly defined anomaly measure.

## 3 Method

### 3.1 The Proposed TAM Approach

**Problem Statement.** We tackle unsupervised anomaly detection on an attributed graph. Specifically, let $G = (\mathcal{V}, \mathcal{E}, \mathbf{X})$ be an attributed graph, where $\mathcal{V} = \{v_1, \cdots, v_N\}$ denotes its node set, $\mathcal{E} \subseteq \mathcal{V} \times \mathcal{V}$ with $e \in \mathcal{E}$ is the edge set, $e_{ij} = 1$ represents there is a connection between node $v_i$ and $v_j$, $\mathbf{X} \in \mathbb{R}^{N \times M}$ denotes the matrix of node attributes and $\mathbf{x}_i \in \mathbb{R}^M$ is the attribute vector of $v_i$, and $\mathbf{A} \in \{0, 1\}^{N \times N}$ is the adjacency matrix of $G$ with $\mathbf{A}_{ij} = 1$ iff $(v_i, v_j) \in \mathcal{E}$, then GAD aims to learn an anomaly scoring function $f : \mathcal{V} \to \mathbb{R}$, such that $f(v) < f(v'), \forall v \in \mathcal{V}_n, v' \in \mathcal{V}_a$, where $\mathcal{V}_n$ and $\mathcal{V}_a$ denotes the set of normal and abnormal nodes, respectively. Per the nature of anomaly, it is typically assumed that $|\mathcal{V}_n| \gg |\mathcal{V}_a|$. But in unsupervised GAD we do not have access to the class labels of the nodes during training. In this work, our problem is to learn an unsupervised local node affinity-based anomaly scoring function $f$.

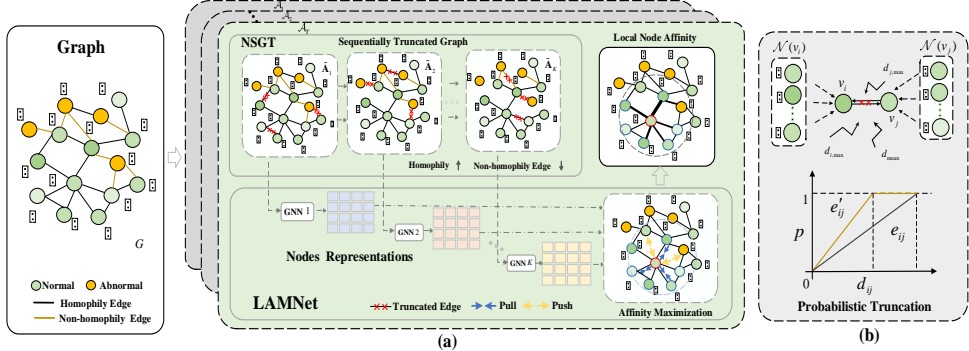

Figure 2: Overview of TAM. (**a**) TAM leverages the observation that normal nodes have stronger affinity relations to their neighbors than anomalies to learn an unsupervised GAD model. It learns a set of affinity maximization GNNs (*i.e.*, LAMNet) on a set of sequentially truncated graphs yielded by our probabilistic graph truncation method NSGT. We build an ensemble of TAM models to make use of the randomness in NSGT for more effective GAD. (**b**) NSGT iteratively removes edges with a probability proportional to the distance between the connected nodes.

**Our Proposed Framework.** Motivated by the one-class homophily phenomenon, we introduce a local node affinity-based anomaly score measure. Since it is difficult to capture and quantify the one-class homophily in the raw attribute space and the generic node representation space, we introduce a truncated affinity maximization (TAM) approach to learn a tailored representation space where the local node affinity can well distinguish normal and abnormal nodes based on the one-class homophily property. As shown in Fig. 2, TAM consists of two novel components, namely *local affinity maximization-based graph neural networks* (LAMNet) and *normal structure-preserved graph truncation* (NSGT). LAMNet trains a graph neural network using a local affinity maximization objective on an iteratively truncated graph structure yielded by NSGT.

NSGT is designed in a way through which we preserve the homophily edges while eliminating non-homophily edges iteratively. The message passing in LAMNet is then performed using the truncated adjacency matrix rather than the original one. In doing so, we reinforce the strong affinity among nodes with homophily relations to their neighbors (*e.g.*, normal nodes), avoiding the potential bias caused by non-homophily edges (*e.g.*, connections to abnormal nodes). The output node affinity in LAMNet is used to define anomaly score, *i.e.*, the weaker the local affinity is, the more likely the node is an abnormal node. There exists some randomness in our NSGT-based graph truncation. We utilize those randomness for more effective GAD by building a bagging ensemble of TAM.

### 3.2 Local Node Affinity as Anomaly Measure

As shown in Fig. 1, normal nodes have significantly stronger homophily relations with each other than the abnormal nodes. However, we do not have class label information to calculate the homophily of each node in unsupervised GAD. We instead utilize the local affinity of each node to its neighbors to exploit this one-class homophily property for unsupervised GAD. The local affinity can be defined as an averaged similarity to the neighboring nodes, and the anomaly score $f$ is opposite to the affinity:

$$h(v_i) = \frac{1}{|\mathcal{N}(v_i)|} \sum_{v_j \in \mathcal{N}(v_i)} \text{sim}(\mathbf{x}_i, \mathbf{x}_j) \; ; \; f(v_i) = -h(v_i), \tag{1}$$

where $\mathcal{N}(v_i)$ is the neighbor set of node $v_i$ and $\text{sim}(\mathbf{x}_i, \mathbf{x}_j) = \frac{\mathbf{x}_i^{\mathrm{T}} \mathbf{x}_j}{\|\mathbf{x}_i\|\|\mathbf{x}_j\|}$ measures the similarity of a node pair $(v_i, v_j)$. The local node affinity $h(v_i)$ is a normal score: the larger the affinity is, the stronger homophily the node has w.r.t. its neighboring nodes based on node attributes, and thus, the more likely the node is a normal node.

The measure in Eq. (1) provides a new perspective to quantify the normality/abnormality of nodes, enabling a much simpler anomaly scoring than existing popular measures such as the reconstruction error $f(v_i) = (1 - \alpha)\|\mathbf{a}_i - \widehat{\mathbf{a}}_i\|_2 + \alpha\|\mathbf{x}_i - \widehat{\mathbf{x}}_i\|_2$, where $\mathbf{a}_i$ denotes the neighborhood structure of $v_i$, $\widehat{\mathbf{a}}_i$ and $\widehat{\mathbf{x}}_i$ are the reconstructed structure and attributes for node $v_i$, and $\alpha$ is a hyperparameter.

Our anomaly measure also provides a new perspective to learn tailored node representations for unsupervised GAD. Instead of minimizing the commonly used data reconstruction errors, based on the one-class homophily, we can learn anomaly-discriminative node representations by maximizing the local node affinity. Our TAM approach is designed to achieve this goal.

## 3.3 TAM: Truncated Affinity Maximization

The local node affinity may not work well in the raw node attribute space since (i) there can be many irrelevant attributes in the original data space and (ii) some abnormal nodes can also be connected to nodes of similar attributes. To tackle this issue, TAM is proposed to learn optimal node representations that maximize the local affinity of nodes that have strong homophily relations with their neighbors in terms of node attributes. Due to the overwhelming presence of normal nodes in a graph, the TAM-based learned node representation space is optimized for normal nodes, enabling stronger local affinity for the normal nodes than the abnormal ones. The TAM-based anomaly scoring using local node affinity can be defined as:

$$f_{TAM}(v_i; \Theta, \mathbf{A}, \mathbf{X}) = -\frac{1}{|\mathcal{N}(v_i)|} \sum_{v_j \in \mathcal{N}(v_i)} \text{sim}(\mathbf{h}_i, \mathbf{h}_j), \tag{2}$$

where $\mathbf{h}_i = \psi(v_i; \Theta, \mathbf{A}, \mathbf{X})$ is a GNN-based node representation of $v_i$ learned by a mapping function $\psi$ parameterized by $\Theta$ in TAM. Below we introduce how we learn the $\psi$ function via the two components of TAM, LAMNet and NSGT.

**Local Affinity Maximization Networks (LAMNet).** LAMNet aims to learn a GNN-based mapping function $\psi$ that maximizes the affinity of nodes with homophily relations to their neighbors, while keeping the affinity of nodes with non-homophily edges are weak. Specifically, the projection from the graph nodes onto new representations using $\ell$ GNN layers can be generally written as

$$\mathbf{H}^{(\ell)} = \text{GNN}\left(\mathbf{A}, \mathbf{H}^{(\ell-1)}; \mathbf{W}^{(\ell)}\right), \tag{3}$$

where $\mathbf{H}^{(\ell)} \in \mathbb{R}^{N \times h^{(l)}}$ and $\mathbf{H}^{(\ell-1)} \in \mathbb{R}^{N \times h^{(l-1)}}$ are the $h^{(l)}$-dimensional and $h^{(l-1)}$-dimensional representations of node $v_i$ in the $(\ell)$-th layer and $(\ell-1)$-th layer, respectively. In the first GNN layer, *i.e.*, when $\ell = 1$, the input $\mathbf{H}^{(0)}$ is set to the raw attribute matrix $\mathbf{X}$. $\mathbf{W}^{(\ell)}$ are the weight parameters of $(\ell)$-th layer. For $\text{GNN}(\cdot)$, multiple types of GNNs can be used [50, 55]. In this work, we employ a GCN (graph convolutional network) [20] due to its high efficiency. Then $\mathbf{H}^{(\ell)}$ can be obtained via

$$\mathbf{H}^{(\ell)} = \phi\left(\mathbf{D}^{-\frac{1}{2}} \mathbf{A} \mathbf{D}^{-\frac{1}{2}} \mathbf{H}^{(\ell-1)} \mathbf{W}^{(\ell-1)}\right), \tag{4}$$

where $\mathbf{D} = \text{diag}(\mathbf{D}_i) = \sum_j \mathbf{A}_{ij}$ is the degree matrix for the graph $G$, and $\phi(\cdot)$ is an activation function. Let $\mathbf{H}^{(\ell)} = \{\mathbf{h}_1, \mathbf{h}_2, ..., \mathbf{h}_N\}$ be the node representations of the last GCN layer, then the mapping function $\psi$ is a sequential mapping of graph convolutions as in Eq. (4), with $\Theta = \{\mathbf{W}^1, \mathbf{W}^2, \cdots, \mathbf{W}^{(\ell)}\}$ be the parameter set in our LAMNet. The following objective can then be used to optimize $\psi$:

$$\min_{\Theta} \sum_{v_i \in \mathcal{V}} \left(f_{TAM}(v_i; \Theta, \mathbf{A}, \mathbf{X}) + \lambda \frac{1}{|\mathcal{V} \setminus \mathcal{N}(v_i)|} \sum_{v_k \in \mathcal{V} \setminus \mathcal{N}(v_i)} \text{sim}(\mathbf{h}_i, \mathbf{h}_k)\right), \tag{5}$$

where the first term is equivalent to maximizing the local affinity of each node based on the learned node representations, while the second term is a regularization term, and $\lambda$ is a regularization hyperparameter. The regularization term adds a constraint that the representation of each node should be dissimilar from that of non-adjacent nodes to enforce that the representations of non-local nodes are distinguishable, while maximizing the similarity of the representations of local nodes.

The optimization using Eq. (5) can be largely biased by non-homophily edges. Below we introduce the NSGT component that helps overcome this issue.

**Normal Structure-preserved Graph Truncation (NSGT).** LAMNet is driven by the one-class homophily property, but its objective and graph convolution operations can be biased by the presence non-homophily edges, *i.e.*, edges that connect normal and abnormal nodes. The NSGT component

is designed to remove these non-homophily edges, yielding a truncated adjacency matrix $\tilde{\mathbf{A}}$ with homophily edge-based normal graph structure. LAMNet is then performed using the truncated adjacency matrix $\tilde{\mathbf{A}}$ rather than the original adjacency matrix $\mathbf{A}$.

Since homophily edges (*i.e.*, edges connecting normal nodes) connect nodes of similar attributes, the distance between the nodes of homophily edges is often substantially smaller than that of non-homophily edges. But there can also exist homophily edges that connect dissimilar normal nodes. These can be observed in GAD datasets, as shown in Fig. 3(a-b) for datasets with synthetic/real anomalies. Motivated by this, NSGT takes a probabilistic approach and performs the graph truncation as follows: for a given edge $e_{ij} = 1$, it is considered as a non-homophily edge and removed (i. e. $e_{ij} = 0$) if and only if the distance between node $v_i$ and node $v_j$ is sufficiently large w.r.t. the neighbor sets of both $v_i$ and $v_j$, $\mathcal{N}(v_i)$ and $\mathcal{N}(v_j)$. Formally, NSGT truncates the graph by

$$e_{ij} \leftarrow 0 \text{ iff } d_{ij} > r_i \ \& \ d_{ij} > r_j, \ \forall e_{ij} = 1, \tag{6}$$

where $d_{ij}$ is a Euclidean distance between $v_i$ and $v_j$ based on node attributes, $r_i$ is a randomly selected value from the range $[d_{mean}, d_{i,max}]$ where $d_{mean} = \frac{1}{m} \sum_{(v_i,v_j)\in\varepsilon} d_{ij}$ is the mean distance of graph, where $m$ is the number of non-zero elements in the adjacent matrix, and $d_{i,max}$ is the maximum distances in $\{d_{ik}, v_k \in \mathcal{N}(v_i)\}$. Similarly, $r_j$ is randomly sampled from the range $[d_{mean}, d_{j,max}]$.

Theoretically, the probability of $r_i < d_{ij}$ can be defined as

$$p\left(r_i < d_{ij}\right) = \frac{\max(d_{ij} - d_{mean}, 0)}{d_{i,\max} - d_{mean}}. \tag{7}$$

Note that this probability is premised on $d_{i,\max} > d_{mean}$, and we set $p\left(r_i < d_{ij}\right) = 0$ if $d_{i,\max} \leq d_{mean}$. Then the probability of $e_{ij}$ being removed during the truncation process is as follows

$$p(\mathcal{E} \setminus e_{ij}) = p\left(r_i < d_{ij}\right) p\left(r_j < d_{ij}\right), \tag{8}$$

As shown in Fig. 2(b), if $e_{ij}$ is a homophily edge, we would have a small $d_{ij}$, which results in small $p\left(r_i < d_{ij}\right)$ and $p\left(r_j < d_{ij}\right)$, and thus, $p(\mathcal{E} \setminus e_{ij})$ is also small. By contrast, a non-homophily edge would result in a large $d_{ij}$, and ultimately a large $p(\mathcal{E} \setminus e_{ij})$. Therefore, NSGT can help eliminate non-homophily edges with a high probability, while preserving the genuine homophily graph structure.

Considering the large difference in the range $[d_{mean}, d_{i,max}]$ for each node and the randomness in truncation, NSGT performs a sequentially iterative truncation rather than a single-pass truncation. In each iteration it randomly removes some non-homophily edges with probability proportional to the distance between their associated nodes, and then it updates the range $[d_{mean}, d_{i,max}]$ for nodes that have edges being removed. Thus, for $K$ iterations on a graph $G$, it would produce a set of $K$ sequentially truncated graphs with the corresponding truncated adjacency matrices $\mathcal{A} = \{\tilde{\mathbf{A}}_1, \tilde{\mathbf{A}}_2, \cdots, \tilde{\mathbf{A}}_K\}$. Note that since the graph is sequentially truncated, we have $\mathcal{E}_{i+1} \subset \mathcal{E}_i$, where $\mathcal{E}_i$ is the edge set left after the i-th iteration. As shown in Fig. 3(c-d), such a sequentially iterative truncation helps largely increase the homophily of the normal nodes, while at the same time effectively removing non-homophily edges gradually.

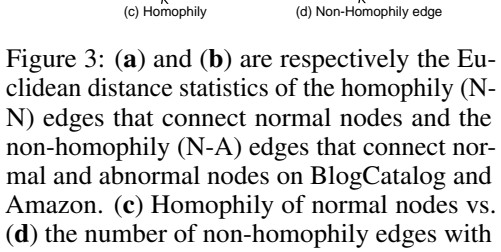

Figure 3: (**a**) and (**b**) are respectively the Euclidean distance statistics of the homophily (N-N) edges that connect normal nodes and the non-homophily (N-A) edges that connect normal and abnormal nodes on BlogCatalog and Amazon. (**c**) Homophily of normal nodes vs. (**d**) the number of non-homophily edges with increasing truncation iterations/depths.

**Training.** To detect anomalies in different graph truncation scales, we train a LAMNet on each of the $K$ sequentially truncated adjacency matrices in $\mathcal{A}$, resulting in $K$ LAMNets parameterized by $\{\Theta_1, \Theta_2, \cdots, \Theta_K\}$ for various truncation depths. Particularly, for each LAMNet, instead of using $\mathbf{A}$, its graph convolutions across all GCN layers are performed on the truncated adjacency matrix $\tilde{\mathbf{A}}_k$ as follows to mitigate the biases caused by the

non-homophily edges in message passing:

$$\mathbf{H}^{(\ell)} = \phi \left( \mathbf{D}^{-\frac{1}{2}} \tilde{\mathbf{A}}_k \mathbf{D}^{-\frac{1}{2}} \mathbf{H}^{(\ell-1)} \mathbf{W}^{(\ell-1)} \right). \tag{9}$$

In doing so, we complete the training of a TAM-based base model, consisting of $K$ LAMNets.

Further, being a probabilistic approach, NSGT has some randomnesses in the graph truncation, and so does the LAMNets. To make use of those randomness, we build an ensemble of TAM models with a size of $T$, as shown in Fig. 2(a). That is, for a graph $G$, we perform NSGT $T$ times independently, resulting in $T$ sets of the truncated adjacency matrix set $\{\mathcal{A}_1, \mathcal{A}_2, \cdots, \mathcal{A}_T\}$, with each $\mathcal{A} = \{\tilde{\mathbf{A}}_1, \tilde{\mathbf{A}}_2, \cdots, \tilde{\mathbf{A}}_K\}$. We then train a LAMNet on each of these $T \times K$ adjacency matrices, obtaining an ensemble of $T$ TAM models (*i.e.*, $T \times K$ LAMNets).

Note that in training the LAMNets, the local affinity in the first term in Eq. (5) is computed using the original adjacency matrix $\mathbf{A}$. This is because we aim to utilize the primary graph structure for local affinity-based GAD; the graph truncation is designed to mitigate the graph convolution biases only.

**Inference.** During inference, we can obtain one local node affinity-based anomaly score per node from each LAMNet, and we aggregate the local affinity scores from all $T \times K$ LAMNets to compute an overall anomaly score as

$$\text{score}(v_i) = \frac{1}{T \times K} \sum_{t=1}^{T} \sum_{k=1}^{K} f_{TAM}(v_i; \Theta_{t,k}^*, \mathbf{A}, \mathbf{X}), \tag{10}$$

where $\Theta_{t,k}^*$ is the learned weight parameters for the LAMNet using the $k$-th truncated adjacency matrix in the $t$-th TAM model. The weaker the local node affinity in the learned representation spaces under various graph truncation scales, the larger the anomaly score the node $v_i$ has.

## 4 Experiments

**Datasets.** We conduct the experiments on six commonly-used publicly-available real-world GAD datasets from diverse online shopping services and social networks, and citation networks, including BlogCatalog [49], ACM [48], Amazon [10], Facebook [56], Reddit, and YelpChi [21]. The first two datasets contain two types of injected anomalies – contextual and structural anomalies [8, 34] – that are nodes with significantly deviated graph structure and node attributes respectively. The other four datasets contain real anomalies. Detailed information about the datasets can be found in App. B.

**Competing Methods and Performance Metrics.** TAM is compared with two state-of-the-art (SOTA) shallow methods – iForest [25] and ANOMALOUS [41] – and five SOTA GNN-based deep methods, including three self-supervised learning based methods – CoLA [28], SL-GAD [63], and HCM-A [17] – and two reconstruction-based methods – DOMINANT [8] and ComGA [31]. iForest works on the raw node attributes, while ANOMALOUS works on the raw node attributes and graph structure. The other methods learn new representation space for GAD.

Following [4, 38, 52, 64], two popular and complementary evaluation metrics for anomaly detection, Area Under the Receiver Operating Characteristic Curve (AUROC) and Area Under the precision-recall curve (AUPRC), are used. Higher AUROC/AUPRC indicates better performance. The reported AUROC and AUPRC results are averaged over 5 runs with different random seeds.

**Implementation Details.** TAM is implemented in Pytorch 1.6.0 with python 3.7 and all the experiments are run on an NVIDIA GeForce RTX 3090 24GB GPU. In TAM, each LAMNet is implemented by a two-layer GCN, and its weight parameters are optimized using Adam [19] optimizer with 500 epochs and a learning rate of $1e-5$ by default. $T = 3$ and $K = 4$ are used for all datasets. Datasets with injected anomalies, such as BlogCatalog and ACM, require strong regularization, so $\lambda = 1$ is used by default; whereas $\lambda = 0$ is used for the four real-world datasets. Hyperparameter analysis w.r.t. $K$ is presented in Sec. 4.2. TAM can perform stably within a range of $T$ and $\lambda$ (see App. C.1 for detail). All the competing methods are based on their publicly-available official source code, and they are trained using their recommended optimization and hyperparameter settings in the original papers.

Table 1: AUROC and AUPRC results on six real-world GAD datasets with injected/real anomalies. The best performance per row is boldfaced, with the second-best underlined.

| Metric | Method | Dataset | | | | | |
|---|---|---|---|---|---|---|---|
| | | BlogCatalog | ACM | Amazon | Facebook | Reddit | YelpChi |
| AUROC | iForest | $0.3765_{\pm0.019}$ | $0.5118_{\pm0.018}$ | $0.5621_{\pm0.008}$ | $0.5382_{\pm0.015}$ | $0.4363_{\pm0.020}$ | $0.4120_{\pm0.040}$ |
| | ANOMALOUS | $0.5652_{\pm0.025}$ | $0.6856_{\pm0.063}$ | $0.4457_{\pm0.003}$ | $\underline{0.9021}_{\pm0.005}$ | $0.5387_{\pm0.012}$ | $\underline{0.4956}_{\pm0.003}$ |
| | DOMINANT | $0.7590_{\pm0.010}$ | $\underline{0.8569}_{\pm0.020}$ | $\underline{0.5996}_{\pm0.004}$ | $0.5677_{\pm0.002}$ | $0.5555_{\pm0.011}$ | $0.4133_{\pm0.010}$ |
| | CoLA | $0.7746_{\pm0.009}$ | $0.8233_{\pm0.001}$ | $0.5898_{\pm0.008}$ | $0.8434_{\pm0.011}$ | $\mathbf{0.6028}_{\pm0.007}$ | $0.4636_{\pm0.001}$ |
| | SL-GAD | $\underline{0.8123}_{\pm0.002}$ | $0.8479_{\pm0.005}$ | $0.5937_{\pm0.011}$ | $0.7936_{\pm0.005}$ | $0.5677_{\pm0.005}$ | $0.3312_{\pm0.035}$ |
| | HCM-A | $0.7980_{\pm0.004}$ | $0.8060_{\pm0.004}$ | $0.3956_{\pm0.014}$ | $0.7387_{\pm0.032}$ | $0.4593_{\pm0.011}$ | $0.4593_{\pm0.005}$ |
| | ComGA | $0.7683_{\pm0.004}$ | $0.8221_{\pm0.025}$ | $0.5895_{\pm0.008}$ | $0.6055_{\pm0.008}$ | $0.5453_{\pm0.003}$ | $0.4391_{\pm0.000}$ |
| | TAM (Ours) | $\mathbf{0.8248}_{\pm0.003}$ | $\mathbf{0.8878}_{\pm0.024}$ | $\mathbf{0.7064}_{\pm0.010}$ | $\mathbf{0.9144}_{\pm0.008}$ | $\underline{0.6023}_{\pm0.004}$ | $\mathbf{0.5643}_{\pm0.007}$ |
| AUPRC | iForest | $0.0423_{\pm0.002}$ | $0.0372_{\pm0.001}$ | $0.1371_{\pm0.002}$ | $0.0316_{\pm0.003}$ | $0.0269_{\pm0.001}$ | $0.0409_{\pm0.000}$ |
| | ANOMALOUS | $0.0652_{\pm0.005}$ | $0.0635_{\pm0.006}$ | $0.0558_{\pm0.001}$ | $0.1898_{\pm0.004}$ | $0.0375_{\pm0.004}$ | $\underline{0.0519}_{\pm0.002}$ |
| | DOMINANT | $0.3102_{\pm0.011}$ | $\underline{0.4402}_{\pm0.036}$ | $\underline{0.1424}_{\pm0.002}$ | $0.0314_{\pm0.041}$ | $0.0356_{\pm0.002}$ | $0.0395_{\pm0.020}$ |
| | CoLA | $0.3270_{\pm0.000}$ | $0.3235_{\pm0.017}$ | $0.0677_{\pm0.017}$ | $\underline{0.2106}_{\pm0.017}$ | $\mathbf{0.0449}_{\pm0.002}$ | $0.0448_{\pm0.002}$ |
| | SL-GAD | $\underline{0.3882}_{\pm0.007}$ | $0.3784_{\pm0.011}$ | $0.0634_{\pm0.005}$ | $0.1316_{\pm0.020}$ | $0.0406_{\pm0.004}$ | $0.0350_{\pm0.000}$ |
| | HCM-A | $0.3139_{\pm0.001}$ | $0.3413_{\pm0.004}$ | $0.0527_{\pm0.015}$ | $0.0713_{\pm0.004}$ | $0.0287_{\pm0.005}$ | $0.0287_{\pm0.012}$ |
| | ComGA | $0.3293_{\pm0.028}$ | $0.2873_{\pm0.012}$ | $0.1153_{\pm0.005}$ | $0.0354_{\pm0.001}$ | $0.0374_{\pm0.001}$ | $0.0423_{\pm0.000}$ |
| | TAM (Ours) | $\mathbf{0.4182}_{\pm0.005}$ | $\mathbf{0.5124}_{\pm0.018}$ | $\mathbf{0.2634}_{\pm0.008}$ | $\mathbf{0.2233}_{\pm0.016}$ | $\underline{0.0446}_{\pm0.001}$ | $\mathbf{0.0778}_{\pm0.009}$ |

## 4.1 Main Results

**Effectiveness on Diverse Real-world Datasets.** The AUROC and AUPRC results on six real-world GAD datasets are reported in Tab. 1. TAM substantially outperforms all seven competing methods on all datasets except Reddit in both metrics, having maximally 9% AUROC and 12% AUPRC improvement over the best-competing methods on the challenge dataset Amazon; on Reddit, it ranks second and performs similarly well to the best contender CoLA. Further, existing methods perform very unstably across different datasets. For example, DOMINANT performs well on ACM but badly on the other datasets; CoLA works well on Reddit but it fails in the other datasets. Similar observations are found in a recent comprehensive comparative study [26]. By contrast, TAM can perform consistently well on these diverse datasets. This is mainly because i) the proposed one-class homophily is more pervasive than the GAD intuitions used by existing methods in different datasets, and ii) TAM offers an effective anomaly scoring function to utilize this anomaly-discriminative property for accurate GAD.

**On Detecting Structural and Contextual Anomalies.** We further examine the effectiveness of TAM on detecting two commonly-studied anomaly types, structural and contextual anomalies, with the results on BlogCatalog and ACM reported in Tab. 2, where the three best competing methods on the two datasets in Tab. 1 are used as baselines. Compared to contextual anomalies, it is significantly more challenging to detect structural anomalies, for which TAM outperforms all three methods in both AUROC and AUPRC. As for contextual anomalies, al-

Table 2: AUROC and AUPRC results of detecting structural and contextual anomalies.

| Metric | Method | Dataset | | | |
|---|---|---|---|---|---|
| | | BlogCatalog | | ACM | |
| | | Structural | Contextual | Structural | Contextual |
| AUROC | DOMINANT | 0.5769 | 0.9591 | 0.6533 | 0.9506 |
| | CoLA | $\underline{0.6524}$ | 0.8867 | $\underline{0.7468}$ | 0.9200 |
| | SL-GAD | 0.5853 | **0.9754** | 0.7354 | **0.9878** |
| | TAM | **0.6819** | $\underline{0.9627}$ | **0.7902** | $\underline{0.9534}$ |
| AUPRC | DOMINANT | $\underline{0.0567}$ | 0.4369 | $\underline{0.0452}$ | 0.5049 |
| | CoLA | 0.0370 | $\underline{0.6298}$ | 0.0381 | $\underline{0.6166}$ |
| | SL-GAD | 0.0359 | 0.4776 | 0.0314 | 0.3083 |
| | TAM | **0.0570** | **0.6308** | **0.0568** | **0.7126** |

though TAM underperforms SL-GAD and ranks in second in AUROC, it obtains substantially better AUPRC than SL-GAD. Note that compared to AUROC that can be biased by low false positives and indicates overoptimistic performance, AUPRC is a more indicative measure focusing on the performance on the anomaly class exclusively [3, 38]. So, achieving the best AUPRC on all four cases of the two datasets demonstrates the superiority of TAM in precision and recall rates for both types of anomaly. These results also indicate that structural anomalies may have stronger local affinity than contextual anomalies, but both of which often have weaker local affinity than normal nodes.

**TAM vs. Raw/Generic Node Representation Space.** As discussed in Sec. 1, local node affinity requires a new node representation space that is learned for the affinity without being biased by non-homophily edges. We provide quantitative supporting results in Tab. 3, where TAM is compared with **Raw Attribute (RA)**, and the spaces learned by DGI [51] and GMI [40]. It is clear that the representations learned by TAM significantly outperforms all three competing representation spaces on all six datasets.

Table 3: Using local node affinity on raw attributes (RA) and learned representation spaces. OOM denotes out of memory on a 24GB GPU.

| Metric | Method | Dataset | | | | | |
|---|---|---|---|---|---|---|---|
| | | BlogCatalog | ACM | Amazon | Facebook | Reddit | YelpChi |
| AUROC | RA | 0.5324 | 0.7520 | 0.6722 | 0.4176 | 0.5794 | 0.3331 |
| | DGI | 0.5647 | 0.7823 | 0.4979 | 0.8647 | 0.5489 | 0.5254 |
| | GMI | 0.5880 | 0.7985 | 0.4438 | 0.8594 | 0.4503 | OOM |
| | TAM (Ours) | **0.8238** | **0.8878** | **0.7064** | **0.9065** | **0.5923** | **0.5541** |
| AUPRC | RA | 0.0652 | 0.1399 | 0.1237 | 0.0193 | **0.0526** | 0.0348 |
| | DGI | 0.0662 | 0.1991 | 0.0719 | 0.1260 | 0.0398 | 0.0551 |
| | GMI | 0.0748 | 0.2251 | 0.0578 | 0.1108 | 0.0281 | OOM |
| | TAM (Ours) | **0.4178** | **0.5124** | **0.2541** | **0.2362** | 0.0446 | **0.0778** |

**Computational Efficiency**. The runtime (including both training and inference time) results are shown in Tab. 4. DOMINANT and ComGA are the simplest GNN-based methods, achieving the most efficient methods. Our method needs to perform multiple graph truncation and train multiple LAMNets, so it takes more time than these two reconstruction-based methods, but it runs much faster than the three recent SOTA models,

Table 4: Runtime (in seconds) results.

| Method | Dataset | | | | | |
| | BlogCatalog | ACM | Amazon | Facebook | Reddit | YelpChi |
|---|---|---|---|---|---|---|
| DOMINANT | 30 | 48 | 9 | 7 | 10 | 26 |
| ComGA | 320 | 542 | 115 | 67 | 159 | 425 |
| HCM-A | 2,664 | 36,254 | 1,111 | 12.7 | 1,228 | 71,891 |
| CoLA | 715 | 2,982 | 2,180 | 84 | 3,800 | 18,194 |
| SL-GAD | 3,055 | 5,728 | 3,071 | 314 | 4,156 | 19,588 |
| TAM (Ours) | 214 | 827 | 362 | 41 | 391 | 837 |

HCM-A, CoLA, and SL-GAD, on most of the datasets. A detailed analysis is provided in App. C.2.

## 4.2 Ablation Study

**Graph Truncation NSGT.** Three alternative approaches to our graph truncation NSGT include:
i) **Raw Graph (RG)** that directly performs affinity maximization on the original graph structure without any graph truncation, and ii) **Edge Drop (ED)** that randomly drops some edges (5% edges by default) [7]. iii) **Similarity Cut (SC)** (removing 5% least similar edges). We compare NSGT to these three approaches in the TAM framework in Tab. 5. It is clear that NSGT consistently and significantly outperforms both

Table 5: NSGT vs. RG and ED.

| Metric | Method | Dataset | | | | | |
| | | BlogCatalog | ACM | Amazon | Facebook | Reddit | YelpChi |
|---|---|---|---|---|---|---|---|
| AUROC | RG | 0.6728 | 0.7511 | 0.4763 | 0.8186 | 0.5575 | 0.4943 |
| | ED | 0.5678 | 0.7162 | 0.4574 | 0.8641 | 0.5641 | 0.5014 |
| | SC | 0.6650 | 0.8668 | 0.5856 | 0.6951 | 0.6007 | 0.4910 |
| | NSGT | **0.8235** | **0.8830** | **0.7120** | **0.9105** | **0.5938** | **0.5449** |
| AUPRC | RG | 0.1849 | 0.1145 | 0.0619 | 0.0808 | 0.0385 | 0.0530 |
| | ED | 0.1229 | 0.1876 | 0.0669 | 0.1204 | 0.0417 | 0.0519 |
| | SC | 0.1621 | 0.5109 | 0.0924 | 0.0410 | **0.0467** | 0.0598 |
| | NSGT | **0.4150** | **0.5152** | **0.2555** | **0.2200** | 0.0449 | **0.0775** |

alternative approaches. *RawGraph* does not work well due to the optimization biases caused non-homophily edges. *EdgeDrop* is ineffective, which is even less effective than *RawGraph*, as it often removes homophily edges rather than the opposite due to the overwhelming presence of such edges in a graph. *SimilarityCut* also significantly underperforms our NSGT on nearly all cases. This is mainly because this variant would fail to take account of local affinity distribution of each node as being captured in NSGT. As a result, it could remove not only non-homophily edges but also homophily edges associated with normal nodes whose local affinity is not as strong as the other normal nodes, which would be the opposite to the objective of the optimization in TAM, leading to less effective detection performance.

**Affinity Maximization Network LAMNet.** The importance of LAMNet is examined by comparing it to its two variants, including **Raw Truncated Affinity (RTA)** that directly calculates the local affinity-based anomaly scores after NSGT (*i.e.*, without involving LAMNet at all), and **DOM** that performs LAMNet but with our affinity maximization objective replaced

Table 6: LAMNet vs. RTA and DOM.

| Metric | Method | Dataset | | | | | |
| | | BlogCatalog | ACM | Amazon | Facebook | Reddit | YelpChi |
|---|---|---|---|---|---|---|---|
| AUROC | RTA | 0.7497 | 0.8043 | 0.6256 | 0.8161 | 0.5783 | 0.5118 |
| | DOM | 0.7642 | 0.8679 | 0.5169 | 0.7793 | 0.5863 | 0.5154 |
| | LAMNet | **0.8248** | **0.8878** | **0.7064** | **0.9144** | **0.5923** | **0.5643** |
| AUPRC | RTA | 0.3329 | 0.2698 | 0.1195 | 0.1212 | 0.0437 | 0.0615 |
| | DOM | 0.3115 | 0.4525 | 0.1517 | 0.1506 | **0.0466** | 0.0538 |
| | LAMNet | **0.4182** | **0.5124** | **0.2630** | **0.2233** | 0.0450 | **0.0766** |

by the popular graph reconstruction loss used in DOMINANT [8]. As shown by the comparison results reported in Tab. 6, LAMNet consistently and significantly outperforms both RTA and DOM, showing that LAMNet can make much better use of the truncated graphs. In addition, by working on our truncated graphs, *DOM* can substantially outperform its original version DOMINANT in Tab. 1 on most of the datasets. This indicates that the proposed one-class homophily property may be also exploited to improve existing GAD methods.

**Anomaly Scoring.** We compare the TAM anomaly scoring to its two variants: i) **TAM-T** that calculates the node affinity in Eq. (5) on the the truncated graph structure rather than the primary graph structure as in TAM, and ii) **Degree** that directly uses the node degree after our graph truncation as anomaly score. As illustrated by the AUPRC results in Fig. 4(a), TAM consistently and significantly outperforms both variants. Degree obtains fairly good performance, which shows that our graph truncation results in structural changes that are

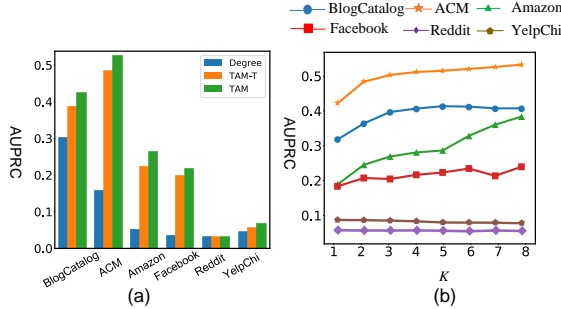

Figure 4: **(a)** TAM vs. Degree and TAM-T. **(b)** TAM results w.r.t. graph truncation depth $K$.

beneficial to GAD. TAM-T underperforms TAM since the local affinity based on the truncated graph structure is affected by randomness and uncertainty in the truncation, leading to unstable and suboptimal optimization. We also show in Fig. 4 (b) that aggregating the anomaly scores obtained from different truncation scales/depths helps largely improve the detection performance. Similar observations are found in the corresponding AUROC results (see App. C.3).

## 4.3 Performance on Large-scale Graphs

In order to demonstrate the effectiveness of TAM on the large-scale datasets, we conduct the experiments on the four large-scale datasets with a large set of nodes and edges, Amazon-all and YelpChi-all by treating the different relations as a single relation following [6], T-Finance [47] and OGB-Proteins [14]. The experimental results are shown in Tab. 7. Due to the increased number of nodes and edges, we set $K = 7$ for these datasets. TAM can perform consistently well on these large-scale datasets and outperforms four comparing methods, which provides further evidence about the effectiveness of our proposed one-class homophily and anomaly measure.

Table 7: Results on large-scale graphs

| Metric | Method | Dataset | | | |
|---|---|---|---|---|---|
| | | Amazon-all | YelpChi-all | T-Finance | OGB-Proteins |
| AUROC | DOMINANT | 0.6937 | 0.5390 | 0.5380 | 0.7267 |
| | ComGA | 0.7154 | 0.5352 | 0.5542 | 0.7134 |
| | CoLA | 0.2614 | 0.4801 | 0.4829 | 0.7142 |
| | SL-GAD | 0.2728 | 0.5551 | 0.4648 | 0.7371 |
| | TAM | **0.8476** | **0.5818** | **0.6175** | **0.7449** |
| AUPRC | DOMINANT | 0.1015 | 0.1638 | 0.0474 | **0.2217** |
| | ComGA | 0.1854 | 0.1658 | 0.0481 | 0.1554 |
| | CoLA | 0.0516 | 0.1361 | 0.0410 | 0.1349 |
| | SL-GAD | 0.0444 | 0.1711 | 0.0386 | 0.1771 |
| | TAM | **0.4346** | **0.1886** | **0.0547** | 0.2173 |

## 4.4 Handling Camouflage Attributes

Fraudsters may adjust their behaviors to camouflage their malicious activities, which could hamper the performance of GNN-based methods. We evaluate the performance of TAM when there are camouflages in the raw attributes. Particularly, we replace 10%, 20%, 30% randomly sampled original attributes with camouflaged attributes, in which the feature/attribute value of the abnormal nodes is replaced (camouflaged) with the mean feature value of the normal nodes. The results are shown in the Tab. 8, which demonstrates that TAM is robust to a high level of camouflaged features and maintains its superiority over the SOTA models that work on the original features. The reason is that NSGT can still successfully remove some non-homophily edges in the local domain, allowing LAMNet to make full use of the one-class homophily and achieve good performance under the camouflage.

Table 8: Results under different levels of camouflage attributes

| Metric | Method | Dataset | | | |
|---|---|---|---|---|---|
| | | 0% | 10% | 20% | 30% |
| AUROC | BlogCatalog | **0.8218** | 0.8045 | 0.8022 | 0.7831 |
| | ACM | **0.8878** | 0.8727 | 0.8688 | 0.8652 |
| | Amazon | **0.7064** | 0.7036 | 0.6954 | 0.6838 |
| | Facebook | **0.9144** | 0.8870 | 0.8804 | 0.8650 |
| | Reddit | **0.6023** | 0.5998 | 0.5915 | 0.5876 |
| | YelpChi | **0.5640** | 0.5447 | 0.5271 | 0.5301 |
| AUPRC | BlogCatalog | **0.4182** | 0.4014 | 0.4024 | 0.4095 |
| | ACM | **0.5124** | 0.4896 | 0.4614 | 0.4822 |
| | Amazon | **0.2634** | 0.2502 | 0.2400 | 0.2324 |
| | Facebook | **0.2233** | 0.2225 | 0.2126 | 0.2015 |
| | Reddit | **0.0446** | 0.0444 | 0.0414 | 0.0440 |
| | YelpChi | **0.0778** | 0.0726 | 0.0701 | 0.0678 |

## 5 Conclusion and Future Work

This paper reveals an important anomaly-discriminative property, the one-class homophily, in GAD datasets with either injected or real anomalies. We utilize this property to introduce a novel unsupervised GAD measure, local node affinity, and further introduce a truncated affinity maximization (TAM) approach that end-to-end optimizes the proposed anomaly measure on truncated adjacency matrix. Extensive experiments on 10 real-world GAD datasets show the superiority of TAM over seven SOTA detectors. We also show that the one-class homophily can be exploited to enhance the existing GAD methods.

**Limitation and Future Work.** TAM cannot directly handle primarily isolated nodes in a graph, though those isolated nodes are clearly abnormal. Additionally, like many GNN-based approaches, including GAD methods, TAM also requires a large memory to perform on graphs with a very large node/edge set. The one-class homophily may not hold for some datasets, such as datasets with strong heterophily relations/subgraphs of normal nodes [1, 29, 36, 46, 62, 66]. Our method would require some adaptations to work well on the dataset with strong heterophily or very large graphs, which are also left for future work.

**Acknowledgments.** We thank the anonymous reviewers for their valuable comments. The work is supported in part by the Singapore Ministry of Education Academic Research Fund Tier 1 grant (21SISSMU031).

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

# A  One-Class Homophily Phenomenon

Fig. 5 shows the one-class homophily phenomenon on the rest of four datasets, in which ACM is a dataset with injected anomalies while the other three datasets contain real anomalies. The results here are consistent with that in the main text.

Given the graph along with the ground truth labels, following [13,33], the homophily and heterophily are defined as the ratio of the edges connecting the node with the same classes and different classes, respectively:

$$
\begin{cases}
X_{\text{hetero}}(v) = \frac{1}{|\mathcal{N}(v)|} |\{u : u \in \mathcal{N}(v), y_u \neq y_v\}| \\
X_{\text{homo}}(v) = \frac{1}{|\mathcal{N}(v)|} |\{u : u \in \mathcal{N}(v), y_u = y_v\}|
\end{cases}
\tag{11}
$$

where $y_v$ is the label to denote whether the node $v$ is an anomaly or not. Note that one-class homophily may not always hold, or it is weak in some datasets. YelpChi which connects the reviews posted by the same user, *e.g.*, YelpCh-RUR is an example of the latter case. As shown in Figure 5, the homophily distribution of normal and abnormal nodes is similar in YelpCh-RUR, whose pattern is different from the large distribution gap in the other three datasets. Since the normal nodes' homophily is generally stronger than the abnormal nodes, we can still successfully remove the non-homophily edges that connect normal and abnormal nodes with a higher probability than the homophily edges, resulting in a truncated graph with stronger one-class homophily.

# B  Description of Datasets

We conduct the experiments on two real-world dataset with injected anomalies and four real-world with genuine anomalies in diverse online shopping services, social networks, and citation networks, including BlogCatalog [49], ACM [48], Amazon [10], Facebook [56], Reddit, YelpChi [21], as well as four large-scale graph datasets. The statistical information including the number of nodes, edge, the dimension of the feature, and the anomalies rate of the datasets can be found in Tab. 9.

Particularly, BlogCatalog is a social blog directory where users can follow each other. Each node represents a user, and each link indicates the following relationships between two users. The attributes of nodes are the tags that describe users and their blogs. ACM is a citation graph dataset where the nodes denote the published papers and the edge denotes the citations relationship between the papers. The attributes of each node are the content of the corresponding paper. BlogCatalog and ACM are popular GAD datasets where the anomalies are injected ones, including structural anomalies and contextual anomalies, which are created following the prior work [28]. Amazon is a graph dataset capturing the relations between users and product reviews. Following [10,60], three different user-user graph datasets are derived from Amazon using different adjacency matrix construction approaches. In this work, we focus on the Amazon-UPU dataset that connects the users who give reviews to at least one same product. The users with less than 20% are treated as anomalies. Facebook [56] is a social network where users build relationships with others and share their same friends. Reddit is a network of forum posts from the social media Reddit, in which the user who has been banned from the platform is annotated as an anomaly. Their post texts were converted to the vector as their attribute. YelpChi includes hotel and restaurant reviews filtered (spam) and recommended (legitimate) by Yelp. Following [35,44], three different graph datasets derived from Yelp using different connections in user, product review text, and time. In this work, we only use YelpChi-RUR which connects reviews posted by the same user. Note that considering it's difficult to conduct an evaluation on the isolated nodes in the graph, they were removed before modeling.

Amazon-all and YelpChi-all [10] are two datasets by treating the different relations as a single relation following [6]. T-Finance [47] is a transaction network. The nodes represent the unique account and the edges represent there are records between two accounts. The attributes of nodes are the features related to registration days, logging activities, and interaction frequency. The anomalies are fraud, money laundering and online gambling which are annotated manually. OGB-Protein [14] is a biological network where node represents proteins and edge indicates the meaningful association between proteins.

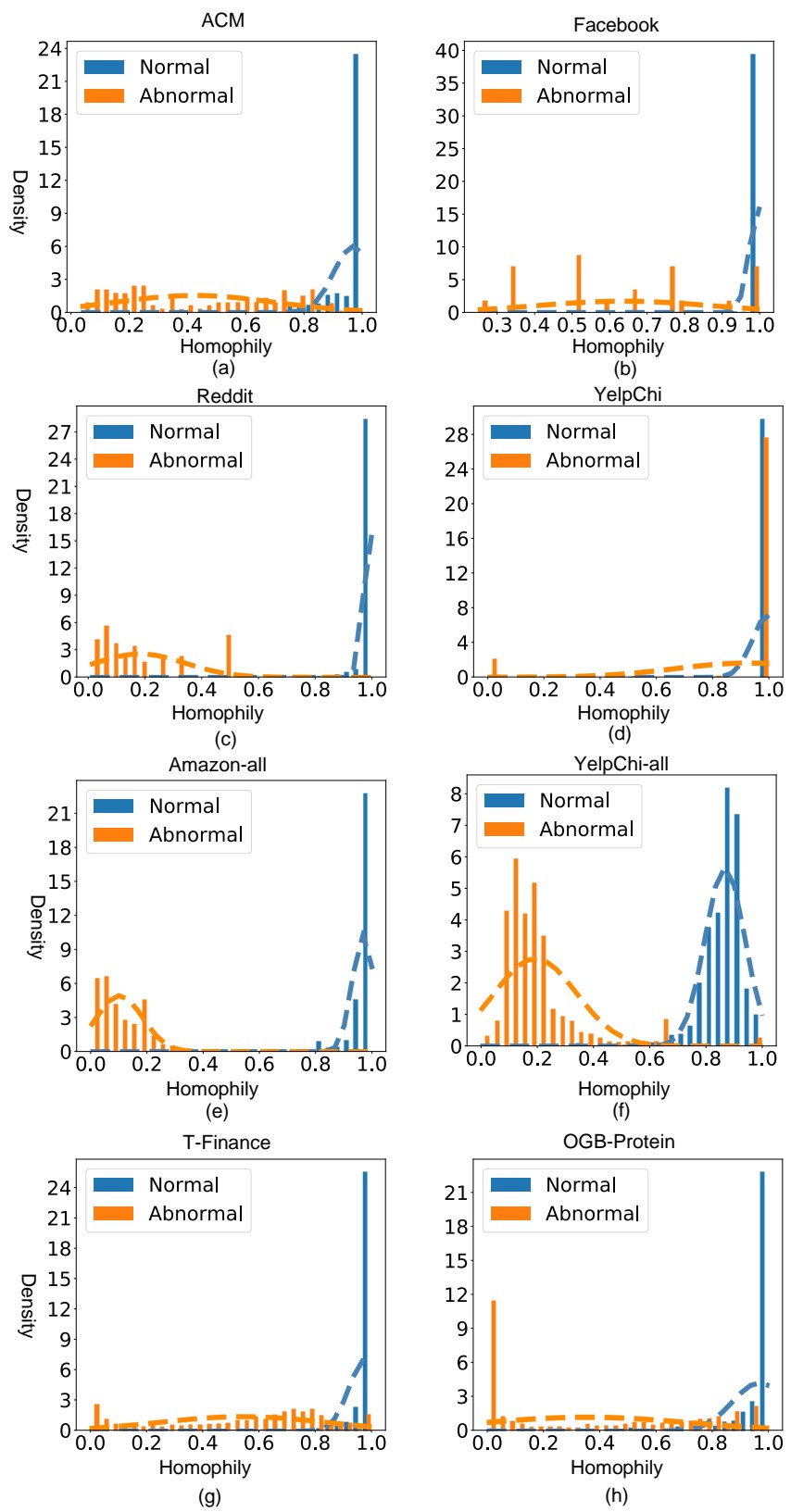

Figure 5: Homophily distribution of normal nodes and abnormal nodes on the rest of eight datasets

Table 9: Key statistics of the datasets. The real-world datasets with Injected/Real anomalies(I/R).

| Data set | Type | R/I | Nodes | Edges | Attributes | Anomalies(Rate) |
|---|---|---|---|---|---|---|
| BlogCatalog | Social Networks | I | 5,196 | 171,743 | 8,189 | 300(5.77%) |
| ACM | Citation Networks | I | 16,484 | 71,980 | 8,337 | 597(3.63%) |
| Amazon | Co-review | R | 10244 | 175,608 | 25 | 693(6.66%) |
| Facebook | Social Networks | R | 4,039 | 88,234 | 576 | 27(0.67%) |
| Reddit | Social Networks | R | 10,984 | 175,608 | 64 | 366(3.33%) |
| YelpChi | Co-review | R | 24,741 | 49,315 | 32 | 1,217(4.91%) |
| Amazon-all | Co-review | R | 11,944 | 4,398,392 | 25 | 821(6.87%) |
| YelpChi-all | Co-review | R | 45,941 | 3,846,979 | 32 | 6,674(14.52%) |
| T-Finance | Transaction Record | R | 39,357 | 21,222,543 | 10 | 1,803 (4.58%) |
| OGB-Protein | Biology Network | I | 132,534 | 39,561,252 | 8 | 6000(4.5%) |

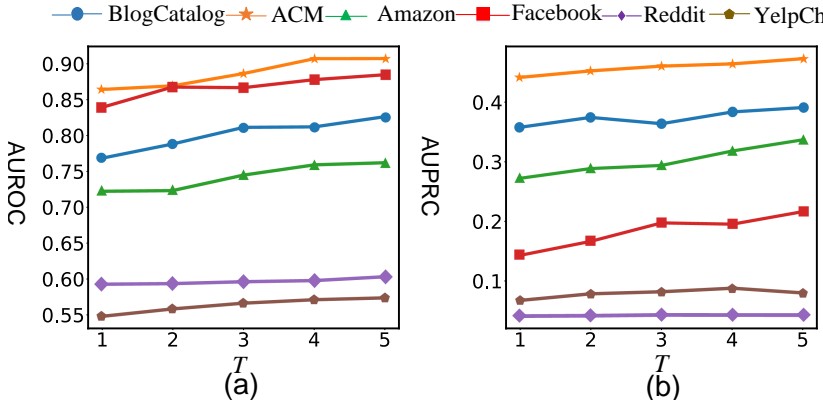

Figure 6: AUROC and AUPRC results w.r.t. ensemble parameter $T$

## C    Additional Experimental Results

### C.1    Hyperparameter Analysis

This section analyzes the sensitivity of TAM w.r.t. two key hyperparameters, including the regularization hyperparameter $\lambda$ and the ensemble parameters $T$. The results on $\lambda$ and $T$ are shown in Fig. 6 and Fig. 7, respectively.

**Ensemble Hyperparameter** $T$. As shown in Fig. 6 with increasing $T$, TAM generally performs better and becomes stable around $T \approx 4$. This is mainly because the use of more ensemble models on truncated graphs reduces the impact of the randomness of truncation and increases the probability of weakening the affinity of abnormal nodes to its neighbors, and this effect would diminish when $T$ is sufficiently large. The average of local affinity from multiple truncated graph sets is more conducive to anomaly detection.

**Regularization Hyperparameter** $\lambda$. In order to evaluate the effectiveness of $\lambda$, we adopt different values to adjust the weight of the regularization. From Fig. 7, we can see that for BlogCatalog and Facebook, adding the regularization improves the effectiveness of the model by a large margin. This is mainly because the use of regularization can prevent all nodes from having identical feature representations. For most real-world datasets with genuine anomalies, the regularization does not significantly improve the effectiveness of the model while decreasing the performance as the increasing of $\lambda$. The main reason is that Amazon, Reddit, and YelpChi are real-world datasets with diverse attributes, and the role of regularization term is not reflected during affinity maximization.

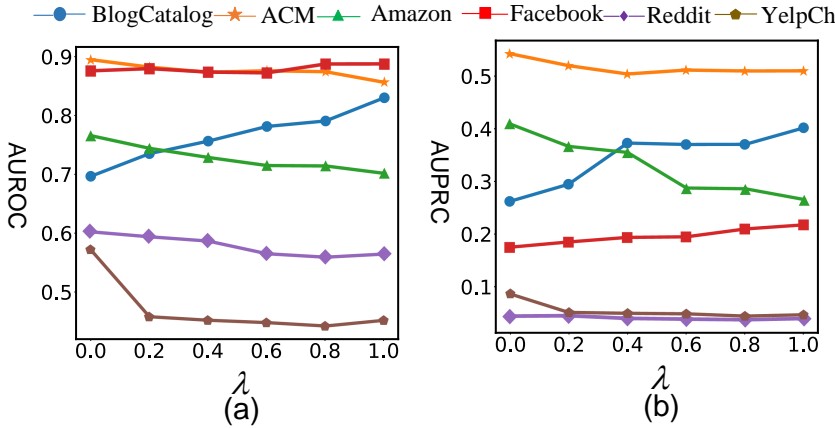

Figure 7: AUROC and AUPRC results w.r.t. regularization hyperparameter $\lambda$

## C.2 Complexity Analysis

This subsection analyzes the time complexity of TAM. Specifically, the distance calculation takes $\mathcal{O}(md_0)$, $m$ is the number of non-zero elements in the adjacent matrix $\mathbf{A}$, and $d_0$ is the dimension of attributes for each node. The graph truncation in TAM takes $2N\eta$, where $N$ is the number of nodes and $\eta$ is the average degree in the graph. In LAMNet, we build a GCN for each truncated graph, which takes $\mathcal{O}(md_1h)$ using sparse-dense matrix multiplications, where $h$ and $d_1$ denotes summation of all feature maps across different layer and feature dimensions in graph convolution operation, respectively. The construction of a GCN takes $\mathcal{O}(md_1h)$. LAMNet also needs to compute all connected pairwise similarities, which takes $\mathcal{O}(N^2)$. Thus, the overall complexity of TAM is $\mathcal{O}(md_0 + (N^2 + md_1h + 2N\eta)KT)$, where $K$ is the truncation depth and $T$ is the ensemble parameter. The complexity is lower than the time complexity in many existing GNN-based graph anomaly detection methods based on the subgraph sampling and hop counting [17, 63].

## C.3 Anomaly Scoring

**AUROC Results of TAM and Its Variants.** We present the AUPRC results of TAM and its two variants, **Degree** and **TAM-T** in the main text, where TAM can consistently and significantly outperform both variants. Similar observation can also be found from the AUROC results in Fig. 8(a). Fig. 8(b) shows the AUROC results of anomaly scoring by aggregating the anomaly scores under all truncation scales/depths. Similar to the AUPRC results in the main text, the anomaly scores obtained from different truncation scales/depths can largely improve the detection performance and the results become stable with increasing graph truncation depth $K$.

**Anomaly Scoring Using Multi-scale Truncation vs Single-scale Truncation.** Fig. 9 shows the results of TAM that performs anomaly scoring based on a single-scale graph truncation rather than the default multi-scale graph truncation. As shown in Fig. 9, the increase of $K$ improves the performance on large datasets such as the Amazon datasets, but it often downgrades the performance on the datasets such as Reddit and YelpChi. The main reason is that the node attributes in these datasets are more similar than the other datasets, restricting the effect of graph truncation. However, the opposite case can occur on the other datasets. To tackle this issue, we define the overall anomaly score as an average score over the anomaly scores obtained from the LAMNets built using truncated graphs under all truncation depths/scales. This resulting multi-scale anomaly score, as shown in Fig. 8(b), performs much more stably than the single-scale anomaly score.

## C.4 Shared-weights LAMNet vs. Non-shared-weights LAMNet

In our experiments, the weight parameters in LAMNets are independent from each other by default, i.e., GNNs in LAMNets are independently trained. In this section, we compare TAM with its variant **TAM-S** where all LAMNets use a single GNN backbone with shared weight parameter.

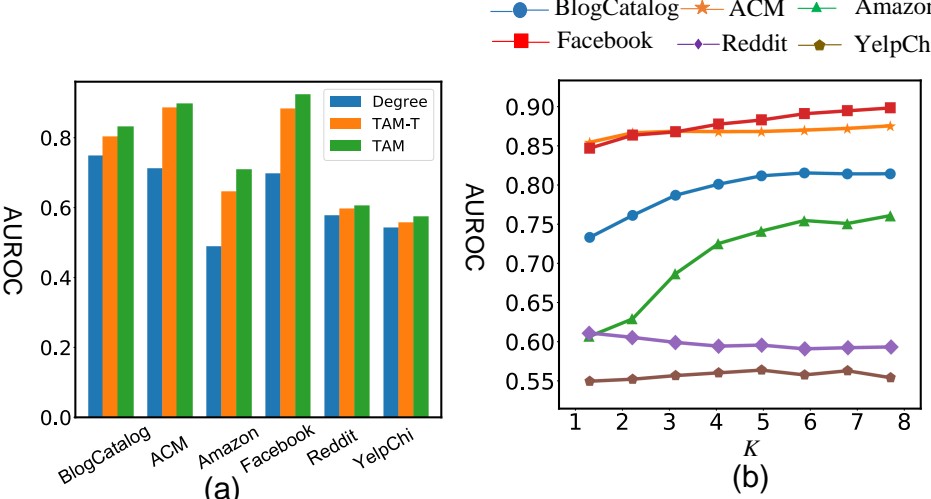

Figure 8: AUROC results of (a) TAM vs. Degree and TAM-T, and (b) TAM w.r.t. graph truncation depth $K$.

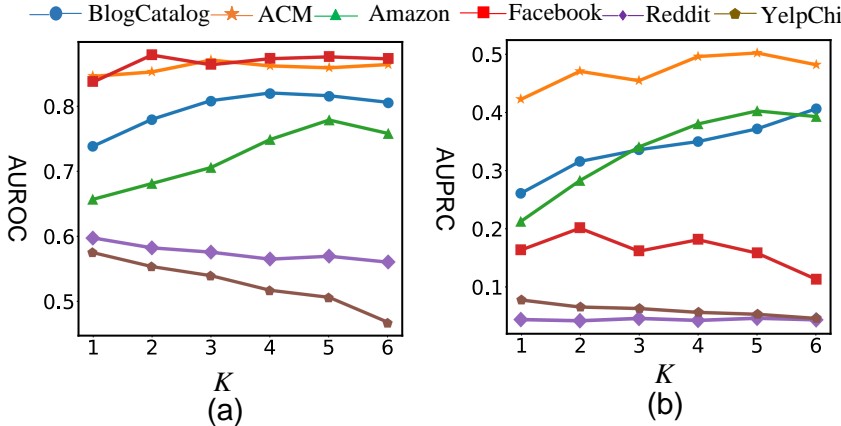

Figure 9: AUROC and AUPRC results of TAM using single-truncation scale-based anomaly scores

The results are shown in Tab. 10. It is clear that TAM performs consistently better than, or comparably well to, TAM-S across the six datasets.

## C.5  Simple Similarity-based Non-homophily Edge Removal

Our ablation study in the main text has shown the inferior performance of *SimilarityCut* method compared to our proposed NSGT, where we directly remove 5% least similar edges. Tab 11 shows the results of removing 10% and 30% least similar edges. It is clear that NSGT outperforms these two other cases of *SimilarityCut*.

## C.6  ROC and Precision-Recall Curves

The ROC curve and Precision-Recall curve of TAM on the six datasets are shown in Fig. 10.

Table 10: AUROC and AUPRC results of TAM using shared-weight LAMNets (TAM-S) vs. non-shared-weight LAMNets (TAM).

| Metric | Method | Dataset | | | | | |
|---|---|---|---|---|---|---|---|
| | | BlogCatalog | ACM | Amazon | Facebook | Reddit | YelpChi |
| AUROC | TAM-S | $0.8170_{\pm 0.002}$ | $0.8826_{\pm 0.003}$ | $0.7044_{\pm 0.008}$ | $\mathbf{0.9165}_{\pm 0.005}$ | $0.6008_{\pm 0.002}$ | $0.5407_{\pm 0.008}$ |
| | TAM | $\mathbf{0.8248}_{\pm 0.003}$ | $\mathbf{0.8878}_{\pm 0.024}$ | $\mathbf{0.7064}_{\pm 0.010}$ | $0.9144_{\pm 0.008}$ | $\mathbf{0.6023}_{\pm 0.004}$ | $\mathbf{0.5643}_{\pm 0.007}$ |
| AUPRC | TAM-S | $0.3908_{\pm 0.002}$ | $0.4960_{\pm 0.001}$ | $0.2597_{\pm 0.002}$ | $0.2087_{\pm 0.006}$ | $\mathbf{0.0459}_{\pm 0.003}$ | $0.0691_{\pm 0.002}$ |
| | TAM | $\mathbf{0.4182}_{\pm 0.005}$ | $\mathbf{0.5124}_{\pm 0.018}$ | $\mathbf{0.2634}_{\pm 0.008}$ | $\mathbf{0.2233}_{\pm 0.016}$ | $0.0446_{\pm 0.001}$ | $\mathbf{0.0778}_{\pm 0.009}$ |

Table 11: Performance under different ratio of similarity-based edge removal. $\theta$ represents the edge-cut ratio based on similarity.

| Metric | Method | Dataset | | | |
|---|---|---|---|---|---|
| | | $\theta=0.05$ | $\theta=0.1$ | $\theta = 0.3$ | TAM |
| AUROC | BlogCatalog | 0.6650 | 0.6526 | 0.6583 | **0.8210** |
| | ACM | 0.8668 | 0.7986 | 0.6911 | **0.8878** |
| | Amazon | 0.5856 | 0.5827 | 0.6106 | **0.7064** |
| | Facebook | 0.6951 | 0.7293 | 0.7934 | **0.9144** |
| | Reddit | 0.6007 | 0.5945 | 0.5758 | **0.6028** |
| | YelpChi | 0.4910 | 0.4872 | 0.4754 | **0.5674** |
| AUPRC | BlogCatalog | 0.1621 | 0.1829 | 0.1729 | **0.4152** |
| | ACM | 0.5109 | 0.5068 | 0.4996 | **0.5124** |
| | Amazon | 0.0924 | 0.1092 | 0.2079 | **0.2634** |
| | Facebook | 0.0410 | 0.1154 | 0.1374 | **0.2233** |
| | Reddit | **0.0467** | 0.0414 | 0.0420 | 0.0446 |
| | YelpChi | 0.0598 | 0.0509 | 0.0519 | **0.0771** |

# D    Description of algorithms

## D.1    Competing Methods

- iForest [25] builds multiple trees to isolate the data based on the node's feature. It has been widely used in outlier detection.

- ANOMALOUS [41] proposes a joint network to conduct the selection of attributes in the CUR decomposition and residual analysis. It can avoid the adverse effects brought by noise.

- DOMINANT [8] leverages the auto-encoder for graph anomaly detection. It consists of an encoder layer and a decoder layer which construct the feature and structure of the graph. The reconstruction errors from the feature and structural module are combined as the anomaly score.

- HCM-A [17] constructs an anomaly indicator by estimating hop count based on both global and local contextual information. It also employs Bayesian learning in predicting the shortest path between node pairs.

- CoLA [28] exploits the local information in a contrastive self-supervised framework. They define the positive pair and negative pair for a target node. The anomaly score is defined as the difference value between its negative and positive score.

- SL-GAD [63] constructs two modules including generative attribute regression and multi-view contrastive for anomaly detection based on CoLA. The anomaly score is generated from the degree of mismatch between the constructed and original features and the discrimination scores.

- ComGA [31] designs a tailor GCN to learn distinguishable node representations by explicitly capturing community structure.

Their implementation is taken directly from their official web pages or the widely-used PyGOD library [26]. The links to the source code pages are as follows:

- iForest: https://github.com/pygod-team/pygod

- ANOMALOUS: https://github.com/pygod-team/pygod

- DOMINANT: https://github.com/kaize0409/GCN_AnomalyDetection_pytorch

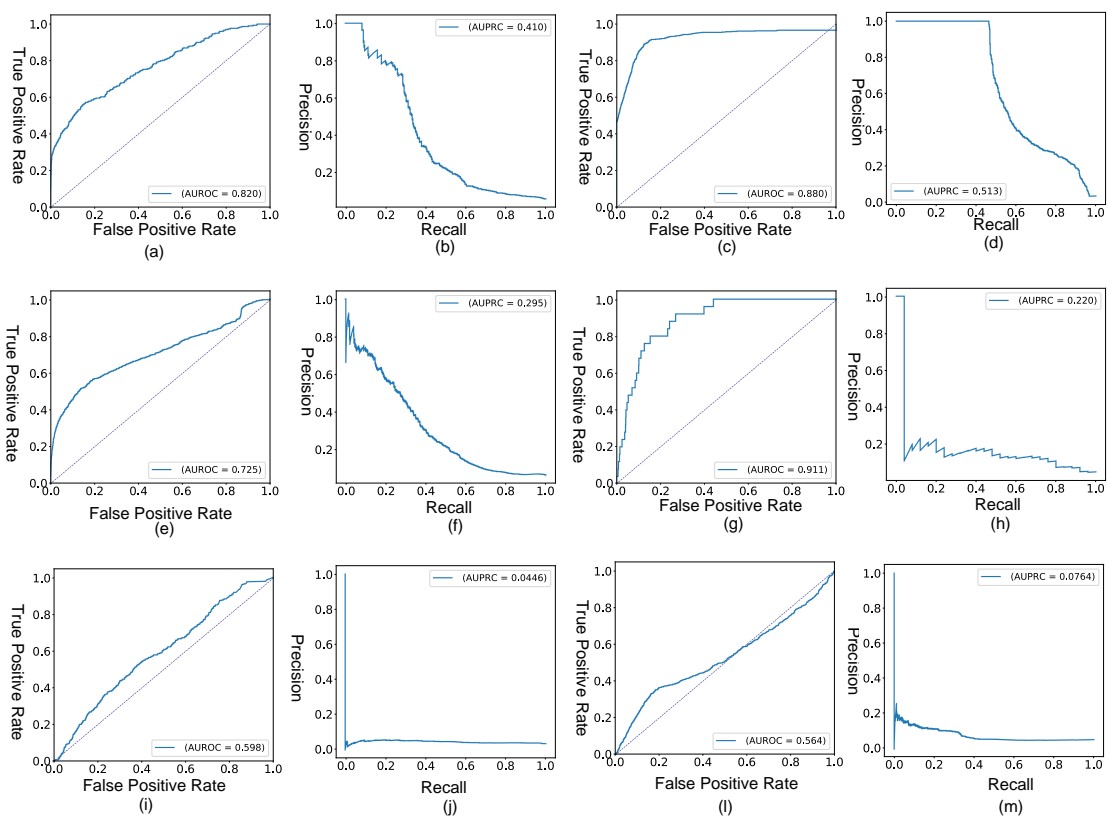

Figure 10: The ROC curve and precision-recall curve of our method TAM on all the six datasets used. (a)(b) BlogCatalog; (c)(d) ACM; (e)(f) Amazon; (h)(i) Facebook; (j)(k) Reddit; (l)(m) YelpChi

- HCM-A: https://github.com/TienjinHuang/GraphAnomalyDetection
- CoLA: https://github.com/GRAND-Lab/CoLA:
- SL-GAD: https://github.com/yixinliu233/SL-GAD
- ComGA:https://github.com/XuexiongLuoMQ/ComGA

## D.2   Pseudo Codes of TAM.

The training algorithms of TAM are summarized in Algorithm 1 and Algorithm 2. Algorithm 1 describes the process of NSGT. Algorithm 2 describes the training process of TAM.

**Algorithm 1** NSGT

    **Input**: Attributed Graph, $G = (\mathcal{V}, \mathcal{E})$, Distance Matrix, $\mathbf{M}$
    **Output**: A truncated graph structure $\tilde{\mathcal{E}}$.

1: /* Regard the graph as a direct graph */
2: Initialize the directed graph truncation indicator $e_{(v_i, v_j)} = 1$ iff $(v_i, v_j) \in \mathcal{E}$
3: Find the mean $d_{mean}$ from $\mathbf{M}$ using $d_{mean} = \frac{1}{m} \sum_{(v_i, v_j) \in \varepsilon} d_{ij}$
4: **for** each $v$ in $\mathcal{V}$ **do**
5:     Find the maximum $d_{v, \max}$ from $\{d(v, v'), (v, v') \in \mathcal{E}\}$
6:     **if** $d_{v, \max} > d_{mean}$ **then**
7:         Randomly sample $r$ from $[d_{mean}, d_{v, \max}]$ for node $v$
8:         **for** each $v'$ in $\{v', (v, v') \in \mathcal{E}\}$ **do**
9:             **if** $d(v, v') > r$ **then**
10:             Plan to cut the edge $v$ to $v'$, i.e., $e_{(v, v')} \leftarrow 0$
11:             **end if**
12:         **end for**
13:     **end if**
14: **end for**
    /* The edge will be removed only when it is a non-homophily edge from both directions of the connected nodes */
15: **for** each $v$ in $\mathcal{V}$ **do**
16:     **for** each $v'$ in $\{v', (v, v') \in \mathcal{E}\}$ **do**
17:         Cut the edge between $v$ and $v'$, $\widetilde{\mathcal{E}} = \mathcal{E} \setminus ((v, v') \cup (v', v))$ ; iff $e_{(v, v')} = e_{(v', v)} = 0$
18:     **end for**
19: **end for**
20: **return** The truncated graph structure $\tilde{\mathcal{E}}$

---

**Algorithm 2** TAM

---

**Input**: Graph, $G = (\mathcal{V}, \mathcal{E}, \mathbf{X})$, $N$: Number of nodes, $L$: Number of layers, $E$: Training epochs, $T$: Ensemble parameter, $K$: Truncation depth.

**Output**: Anomaly scores of all nodes $s$.

1: Compute the Euclidean distance $\mathbf{M} = \{d_{ij}\}$ for each connected node pair $(v_i, v_j) \in \mathcal{E}$
2: Randomly initialize GNN $(\mathbf{h}_1^{(0)}, \mathbf{h}_2^{(0)}, ..., \mathbf{h}_N^{(0)}) \leftarrow \mathbf{X}$ , $\mathcal{E}^{(1,0)}, , ..., \mathcal{E}^{(T,0)} \leftarrow \mathcal{E}$
3: **for** $k = 1, \cdots, K$ **do**
4:    **for** $t = 1, \cdots, T$ **do**
5:       /* Graph truncation and update the graph structure */
6:       $\mathcal{E}^{(t,k)} = \text{NSGT}(\mathcal{V}, \mathcal{E}^{(t,k-1)}, \mathbf{M})$
      /* LAMNet */
7:       **for** $epoch = 1, \cdots, E$ **do**
8:          **for** each $v$ in $\mathcal{V}$ **do**
9:             **for** $l = 1, \cdots, L$ **do**
10:               $\mathbf{h}_{v,l} = \phi(\mathbf{h}_{v,l-1}; \Theta_{t,k})$
11:               $\mathbf{h}_{v,l} = \text{ReLU}\left(\text{AGG}(\{\mathbf{h}_{v',l} : (v, v') \in \mathcal{E}^{(t,k)}\})\right)$
12:             **end for**
13:             Calculate $f_{TAM}(v_i; \Theta_{t,k}, \mathbf{A}, \mathbf{X})$ by Eq. (2).
14:          **end for**
15:          /* Affinity maximization */
16:          Minimize $\sum\limits_{v_i \in \mathcal{V}} \left( f_{TAM}(v_i; \Theta_{t,k}, \mathbf{A}, \mathbf{X}) + \lambda \frac{1}{|\mathcal{V} \backslash \mathcal{N}(v_i)|} \sum\limits_{v_k \in \mathcal{V} \backslash \mathcal{N}(v_i)} \text{sim}(\mathbf{h}_i, \mathbf{h}_k) \right)$
17:          Update $\Theta_{t,k}$ by using stochastic gradient descent
18:       **end for**
19:    **end for**
20: **end for**
   /* Aggregated anomaly score over $T$ sets of multi-scale graph truncation depths */
21: **return** Anomaly Score by $s(v) = \frac{1}{T \times K} \sum\limits_{k=1}^{K} \sum\limits_{t=1}^{T} f_{TAM}(v_i; \Theta_{t,k}^*, \mathbf{A}, \mathbf{X}))$

---

