# OpenReview forum: "Truncated Affinity Maximization: One-class Homophily Modeling for Graph Anomaly Detection"
_NeurIPS.cc/2023/Conference — NeurIPS 2023 poster_

### Official Review · Reviewer_oFzr · 2023-06-20

**Soundness:** 3 good
**Presentation:** 3 good
**Contribution:** 2 fair
**Rating:** 5
**Confidence:** 3

**Summary:**

This paper proposed an unsupervised algorithm for detecting abnormal nodes in a graph. Based on the one-class homophily assumption and the overwhelming presence of normal nodes in a graph, this paper uses the truncated affinity maximization to enable stronger local affinity for normal nodes than abnormal ones.  To eliminate the training bias brought by the non-homophily edges, Normal Structure-preserved Graph Truncation is proposed to remove non-homophily edges iteratively. The proposed method achieves competitive performance on datasets with synthetic and real anomalies.

**Strengths:**

1. The paper is well-structured and easy to follow.
2. The proposed method TAM has strong motivation. The unsupervised setting is close to realistic scenario that normal nodes and anomalies exist in a graph and their labels are unknown.
3. Extensive experiments prove that TAM is a strong baseline for unsupervised anomaly node detection


**Weaknesses:**

1. I suggest adding some supervised graph anomaly detection method on section 2, such as CARE-GNN [1], PC-GNN [2] and Fraudre [3]. Although the supervised setting is far from reality, these algorithms also proposed some ideas for eliminating the negative impact of heterophily edges on anomaly detection.
2. Is TAM wide-applicable on different GNNs? How does TAM perform on other GNN backbones other than GCN?
3. In table 1, why baselines and TAM perform so poor regarding AUPRC. It would be better to draw the roc curve and precision-recall curve.
4. It is difficult to understand Figure 2 (b), what are $c_1$, $c_2$, $c_k$? I suggest reorganizing this illustration.
5. I suggest placing the limitation in a separate paragraph or subsection. Besides the limitations discussed in Conclusion, are there any other shortcoming of TAM?

[1] Enhancing graph neural network-based fraud detectors against camouflaged fraudsters

[2] Pick and choose: a GNN-based imbalanced learning approach for fraud detection

[3] Fraudre: Fraud detection dual-resistant to graph inconsistency and imbalance


**Questions:**

Please refer to Weaknesses.

**Limitations:**

yes

---

> ### Author Rebuttal · Authors · 2023-08-09
>
> Thank you very much for the overall positive rating and constructive comments. We are grateful for the positive comments on our paper clarity, research motivation, and empirical justification. Please see our response to your comments one-by-one below.
>
> > **Weaknesses #1**: To add some supervised graph anomaly detection methods in Sec. 2
>
> Thank you very much for the suggestion.  Supervised graph anomaly detection has drawn increasing attention in recent years, such as the emerging of multiple supervised methods like CARE-GNN,  PCGNN, Fraudre , BWGNN , and GHRN. These methods propose various supervised effective methods to deal with heterophily edges in GNN aggregation, so as to better improve the performance of anomaly detection. We agree that despite being supervised methods, they may inspire designs on exploring the one-class homophily property and alike for better unsupervised graph anomaly detection. We will discuss the corresponding references to discuss those supervised methods and possible opportunities along this research line in Sec. 2.
>
> > **Weaknesses #2**: How does TAM perform on other GNN backbones other than GCN?
>
> As shown by the results in Tables A1 and A2 below, TAM can also work effectively when using GNN backbones other than GCN.  Particularly, TAM using GAT can have some large improvement on multiple datasets like Amazon and Facebook than GCN, while having similar performance on the other datasets. GraphSage and GIN perform inferior to GCN on most of the datasets, since using backbones like GraphSage would lead to the loss of important local information of some nodes due to its small node sampling size during neighborhood aggregation. The inferior effect of GIN might be because our task is different from supervised node classification, where we emphasize more on how to learn the representation of nodes while our method emphasizes more on learning the local node affinity. In general, TAM can achieve good performance using other GNN backbones, but it can have large performance drop on some datasets when comparing to the performance using GCN/GAT, so TAM works best using GCN or GAT.
>
> ```
> Table A1. AUC-ROC for Different GNN Backbones.
> ```
> |**Data**|**GCN**|**GAT**|**GraphSage**|**GIN**|
> |:----|:----:|:----:|:----:|:----:|
> |BlogCatalog|**0.8248**|0.8145| 0.7767|0.7682|
> |ACM|0.8878|**0.8891**|0.7313|0.7833|
> |Amazon|0.7064|0.7245|**0.7532**|0.7034|
> |Facebook |0.9144|**0.9394**|0.7372|0.8216|
> |Reddit   |**0.6023**|0.5999|0.5629|0.5348|
> |YelpChi|0.5643|**0.5787**|0.5243|0.5614|
>
> ```
> Table A2. AUC-PR for Different GNN Backbones.
> ```
> |**Data**|**GCN**|**GAT**|**GraphSage**|**GIN**|
> |:----|:----:|:----:|:----:|:----:|
> |BlogCatalog|**0.4182**|0.3952|0.3844|0.3615|
> |ACM|0.5124|**0.6701**|0.4940|0.4087|
> |Amazon|0.2634|0.3127|**0.4407**|0.2666|
> |Facebook |0.2233|**0.3010**|0.1384|0.1578|
> |Reddit   |**0.0446**|0.0407|0.0409|0.0377|
> |YelpChi|0.0778|**0.0778**|0.0665|0.0727|
>
>
> > **Weaknesses #3**: In table 1, why baselines and TAM perform so poor regarding AUPRC. To draw the roc curve and precision-recall curve.
>
> Thank you very much for the question and suggestion. Since 1) anomaly detection datasets are often an extremely imbalanced one and 2) no labeled data is available for training, it is very difficult for unsupervised detectors to achieve both high precision and recall rates on detecting the rare anomalies, leading to small values of AUC-PR (or AUPRC). As for AUC-ROC, its performance is often overoptimistic because 1) it is affected by performance on both normal and anomaly classes and 2) the normal class is the dominant class whose performance would bias the overall performance.
>
> As a result, it is common to achieve pretty good AUC-ROC but poor AUC-PR performance in unsupervised anomaly detection task (see the results of unsupervised detectors on the far left in Figures D4 & D5 in *Han, S., Hu, X., Huang, H., Jiang, M., & Zhao, Y. (2022). ADBench: Anomaly Detection Benchmark* for some evidence), except that the test data contains many anomalies .
> AUC-PR and AUC-ROC are commonly used as complementary evaluation metrics in graph anomaly detection in the community, because AUC-PR reflects more realistic performance on the anomaly class while AUC-ROC has a good probabilistic interpretation.
>
> We have inserted the ROC curve and Precision-Recall curve of TAM on the six datasets used in Figure 2 in the pdf file in the Author Rebuttal section above. We will add them in the final paper.
>
> > **Weaknesses #4**: Difficult to understand $c_1$,  $c_2$,  and $c_k$ in Figure 2 (b).
>
> Thank you very much for the question and suggestion. We originally intended to use $c_1$,  $c_2$,  and $c_k$ to represent the progressive iteration of our graph truncation. We will reorganize this illustration to provide a more straightforward view of the iterative graph truncation.
>
> > **Weaknesses #5**: Are there any other shortcoming of TAM?
>
> Thank you very much for the suggestion and question. We will place the discussion of TAM's limitations in a separate paragraph.
> In addition to the possible presence of non-homophily cases in some real-world datasets, as also pointed out by Reviewer **NSnj**, another potential limitation is that TAM cannot directly handle primarily isolated nodes in a graph, though those isolated nodes are clearly abnormal if they are rare and the other nodes are connected to at least some nodes. Additionally, like many GNN-based approaches, including graph anomaly detection methods, TAM also requires a large memory to perform on graphs with a very large node/edge set.
>
> Note that these limitations do not affect the detection effectiveness of TAM on a variety of popular real-world graph anomaly detection datasets. Most importantly, it is a seminal work on the idea of one-class homophily and local node affinity maximization specifically for graph anomaly detection, which opens up some great opportunities for exploring substantially more effective anomaly detectors from a new perspective.

---

### Official Review · Reviewer_zqNH · 2023-07-03

**Soundness:** 3 good
**Presentation:** 3 good
**Contribution:** 3 good
**Rating:** 6
**Confidence:** 4

**Summary:**

This paper studies the problem of graph anomaly detection and the authors proposed a novel method based on an identified property named one-class homophily. It is observed that normal nodes have strong connections with each other while abnormal nodes have weaker connections. Existing GAD methods overlook this property. The authors propose a novel unsupervised anomaly scoring measure, i.e., local node affinity, that considers the similarity of nodes to their neighbors. They introduce Truncated Affinity Maximization (TAM) to learn tailored node representations for this measure. TAM optimizes on truncated graphs to mitigate bias from non-homophily edges. Experimental results on several real-world graphs with both manually injected and real anomalies show that TAM outperforms several competing models, achieving over 10% improvement in AUROC/AUPRC on challenging datasets.

**Strengths:**

+ This paper focuses on an interesting and important problem. Graph anomaly detection is of value in both research and practice, especially since it is beneficial for various applications.

+ The proposed method is technically sound. The performance in several data with both manually injected and real-world anomalies is promising.

+ The experimental results show that the proposed method achieves consistent performance improvements over the baselines, which demonstrates its superiority. Also, a comprehensive ablation study to show the effectiveness of different components.

**Weaknesses:**

- There is no theoretical analysis to justify the rationale of the proposed method.

- The proposed method has not been tested on large-scale graphs to show its efficiency.

- The design of some components may need more investigation empirically (see details below).

**Questions:**

- Theoretical analysis. Although the proposed method has shown its effectiveness using comprehensive experiments, i.e., comparing seven SOTA on real-world graphs with both manually injected and real anomalies, there is no theoretical analysis to justify the rationale of the proposed method.

- Efficiency test. The largest graph used in the experiments contains ~4k nodes and ~175k edges, which are relatively not very large. To better demonstrate the efficiency of the method, larger graph benchmarks may be needed, e.g., OGB.

- Component design and impact. Some components may need more investigation empirically, especially the graph truncation strategy as well as the impact of different K. In detail, (1) in section 4.2, the raw graph and (random) edge drop have been compared. A straightforward question is what is the performance of simple similarity-based methods, e.g., removing some less similar edges? (2) Since there are K truncated graphs and each one relies on the previous one, the number of edges will be less and less. Is it possible that for larger K, the graph will contain quite some isolated small subgraphs? If so, is there any negative impact on the performance?

Some minor comments:
- Figure 4(b) exhibits a variation in performance trends across different graphs. Particularly, on the Reddit and YelpChi datasets, the AUPRC decreases as the value of K increases. Is there an explanation for this contrasting behavior?

**Limitations:**

No potential negative societal impact or the authors adequately addressed the limitations.

---

> ### Author Rebuttal · Authors · 2023-08-10
>
> Thank you very much for the overall positive rating and constructive comments. We are grateful for the positive comments on our studied problem, technical contribution and empirical justification. Please see our detailed response below.
>
> > **Weaknesses/Questions #1**: Lack of theoretical analysis to justify our method.
>
> Thank you very much for the comments. We, for the first time, reveal the one-class homophily property in graph anomaly detection datasets by a number of empirical justifications. We further introduce a novel graph anomaly detection approach that provides a principled framework to leverage this property for the task. Unlike the overwhelming reconstruction-based and self-supervised learning frameworks, our approach offers a fundamentally different perspective on designing effective graph anomaly detection. Thus, although there is a lack of theoretical analysis, our work presents some solid findings and interesting insights into the graph anomaly detection problem. Therefore, while we're still working on an in-depth theoretical analysis of the proposed property and TAM, it would be beneficial to the anomaly detection community by publishing these interesting findings and novel insights.
>
>
> > **Weaknesses/Questions #2**: Results on large-scale graph datasets.
>
> Please refer to our reply to **Global Response to Shared Concern #2** in the overall Author Rebuttal section above for detailed discussions on this point.
>
> > **Weaknesses/Questions #3**: The design of some components may need more investigation empirically (see details below), e.g., (1) simple similarity-based graph truncation methods, and (2) the impact of larger $K$ and possibly isolated small subgraphs.
>
> **Re: (1) Simple similarity-based non-homophily edge removal**. We agree that we may perform a deterministic similarity-based removal of less similar edges to obtain the truncated graph, and then optimize LAMNet using such a truncated graph. We present the experimental results of this TAM variant in Tables A1 and A2 below using varying similarity threshold $\theta$. It is clear that this TAM variant significantly underperforms our default TAM on most of the datasets, i.e., BlogCatalog, Amazon, Facebook, and YelpChi. This is mainly because this variant would fail to take account of local affinity distribution of each node as being captured in NSGT. As a result, it could remove not only non-homophily edges but also homophily edges associated with normal nodes whose local affinity is not as strong as the other normal nodes, which would be the opposite to the objective of the optimization in our approach TAM, leading to less effective detection performance.
> Thus, the simple similarity-based method cannot serve as an effective alternative to NSGT.
>
> ```
> Table A1. AUC-ROC Results of Using Similarity-based Edge Removal.
> ```
> |**Data**|**BlogCatalog**|**ACM**|**Amazon**|**Facebook**|**Reddit**|**YelpChi**|
> |:----|:----:|:----:|:----:|:----:|:----:|:----:|
> |$\theta$=0.05|0.6650|0.8668|0.5856|0.6951|0.6007|0.4910|
> |$\theta$=0.1|0.6526|0.7986|0.5827|0.7293|0.5945|0.4872|
> |$\theta$=0.3|0.6583|0.6911|0.6106|0.7934|0.5758|0.4754|
> |TAM|**0.8210**|**0.8878**|**0.7064**|**0.9144**|**0.6028**|**0.5674**|
>
> ```
> Table A2. AUC-PR Results of Using Similarity-based Edge Removal.
> ```
> |**Data**|**BlogCatalog**|**ACM**|**Amazon**|**Facebook**|**Reddit**|**YelpChi**|
> |:----|:----:|:----:|:----:|:----:|:----:|:----:|
> |$\theta$=0.05|0.1621|0.5109|0.0924|0.0410|**0.0467**|0.0598|
> |$\theta$=0.1|0.1829|0.5068|0.1092|0.1154|0.0414|0.0509|
> |$\theta$=0.3|0.1729|0.4996|0.2079|0.1374|0.0420|0.0519|
> |TAM|**0.4152**|**0.5124**|**0.2634**|**0.2233**|0.0446|**0.0771**|
>
> **Re: (2) the impact with increasing truncation steps K?** A short answer is: No. Specifically, in the process of early graph truncation, the edge will gradually decrease. However, since our truncation is based on the average similarity of the entire graph, all edges would stabilize near the average similarity after some truncation iterations. The changes in the graph also gradually stabilize. Since we do not have access to class labels, it is difficult to evaluate whether the truncation is too aggressive. Given the fact that the number of abnormal nodes per graph is assumed to be small, and so does the number of non-homophily edges, and thus, a small $K$, $K=5$, is used by default in our experiments.
>
> **Re: (2) any isolated small subgraphs produced by NSGT and their potential impact?** Yes, it is possible. If we perform multiple graph truncations, there would be some isolated small subgraphs, but this does not affect our subsequent affinity maximization. This is because NSGT is proposed to mitigate the bias toward non-homophily edges in the GNN neighborhood aggregation. This would only affect the neighborhood aggregation in GNNs; the calculation of the local node affinity-based anomaly score is still based on the original graph structure. Further, the small subgraphs resulting from NSGT indicate many nodes of the subgraphs have small affinity to the remaining nodes, indicating some sort of weak affinity of the nodes in the subgraphs. As a result, the local node affinity-based anomaly score for these subgraph nodes would be large, and furthermore, given the weak affinity of these subgraph nodes, their anomaly scores would also be large even if the subgraphs are not isolated after NSGT.
>
> > **Minor Comments**: a variation in the performance on Reddit and YelpChi in Figure 4(b).
>
> Thank you very much for the question. Being a probabilistic method,  the graph truncation has the risk of removing some homophily edges with relatively large distances. This risk can increase with increasing K values, as shown in Figure 5, Appendix C.3. We have considered this situation and addressed this potential issue by (1) adopting an ensemble-based scoring method that aggregates the anomaly scores across different K values to obtain stabilize the detection performance and (2) using a relatively small K value.

---

> > ### Comment · Reviewer_zqNH · 2023-08-18
> > **Thanks for the rebuttal**
> >
> > Thanks for the rebuttal especially the added results. I read the authors' responses as well as other reviews and responses. Most of my concerns have been addressed. I would like to increase my rating.

---

> > > ### Author Response · Authors · 2023-08-19
> > > **Thanks for the increased rating**
> > >
> > > It's great that our rebuttal has addressed most of your concerns. Thank you very much for the increased rating and your support on our work!

---

### Official Review · Reviewer_uzaW · 2023-07-07

**Soundness:** 3 good
**Presentation:** 3 good
**Contribution:** 3 good
**Rating:** 7
**Confidence:** 5

**Summary:**

In this paper, the authors introduced a novel unsupervised anomaly scoring measure (local node affinity) for GAD, and further proposed a Truncated Affinity Maximization (TAM) for GAD. TAM learns tailored node representations for our anomaly measure by maximizing the local affinity of nodes to their neighbors, and is optimized on truncated graphs where non-homophily edges are removed iteratively to mitigate this bias. The authors also conduct a series of experiments to evaluate the anomaly detection performance of TAM.

**Strengths:**

1. This paper is well-motivated and easy to follow.
2. The idea of exploring the one-class homophily characteristic for designing a GAD method is interesting.
3. The experimental section provides comprehensive evaluations of the proposed method from different perspectives, and demonstrates the effectiveness of the proposed method.

**Weaknesses:**

1. In the related work section, the authors should discuss the connections and differences between the proposed methods and other existing works, as well as the necessity for this work.
2. As the authors claimed, one-class homogeneity does not hold in all cases. Therefore algorithms designed on this basis would be relatively limited.
3. The authors propose to optimize TAM on truncated graphs to avoid the negative impact of non-homophily edges. How the author guarantees to obtain ideal truncated graphs with the removal of non-homophily edges iteratively. Besides, does it have some criteria to judge whether the truncation map obtained is ideal or not?

**Questions:**

See weakness.

**Limitations:**

See weakness.

---

> ### Author Rebuttal · Authors · 2023-08-09
>
> Thank you very much for the overall positive rating and constructive comments. We are grateful for the positive comments on our research motivation, readability, our technical design and empirical justification. Please see our response to your comments one-by-one below.
>
> > **Weaknesses #1**: In the related work section, the authors should discuss the connections and differences between the proposed methods and other existing works, as well as the necessity for this work.
>
> Thank you very much for the comment and suggestion. We will add discussions on the connections and differences w.r.t. existing studies in the Related Work section in the final version of the paper.
>
> Specifically, current unsupervised graph anomaly detection methods can be generally grouped into reconstruction-based and self-supervised-based approaches.
>
> Being GNN-based approaches, both the reconstruction-based approaches and our approach TAM rely on node's neighborhood information to calculate the anomaly scores, but we explicitly define an anomaly measure from a new perspective, i.e., local node affinity. This results in fundamentally different anomaly detection optimization objectives. Some of these points have been elaborated in detail in Section 3.2. We will include this discussion into Related Work to brief the connections and the differences.
>
> Similar to the self-supervised approaches, the optimization of TAM also relies on an unsupervised objective. However, the self-supervised approaches require the use of some pre-text tasks like surrogate contrastive learning or classification tasks to learn the feature representations for anomaly detection. By contrast, the optimization of TAM is directly driven by a new, plausible anomaly measure, which enables end-to-end optimization of an explicitly defined anomaly scoring measure.
>
> Overall, TAM offers a very different perspective on devising graph anomaly detection methods, which is tasked to learn tailed node representations for an explicitly defined graph anomaly measure; whereas both reconstruction-based and self-supervised-based approaches are focused more on learning latent node feature representations for a proxy task. Thus, being driven by a prevalent one-class homophily property, TAM and its future variants are expected to align better with the graph anomaly detection problem.
>
> > **Weaknesses #2**: As the authors claimed, one-class homogeneity does not hold in all cases. Therefore algorithms designed on this basis would be relatively limited.
>
> Thank you very much for the comment. Please refer to our reply to **Global Response to Shared Concern #1** in the overall Author Rebuttal section above for a detailed discussion on this point.
>
> > **Weaknesses #3**: The authors propose to optimize TAM on truncated graphs to avoid the negative impact of non-homophily edges. How the author guarantees to obtain ideal truncated graphs with the removal of non-homophily edges iteratively. Besides, does it have some criteria to judge whether the truncation map obtained is ideal or not?
>
> Thank you very much for the comment and the question. Since it is unsupervised anomaly detection, no ground truth is given to evaluate/prove whether a resulting truncated graph is better than another one. Instead we introduce a plausible probabilistic method that helps iteratively eliminate non-homophily edges with a high probability, while preserving the genuine homophily, as elaborated in the NSGT subsection in the paper. In addition, we utilize an ensemble-based anomaly scoring method that aggregates the anomaly scores obtained from a set of multiple graph truncation scales, which enables TAM to 1) better utilize the dominant homophily edges for anomaly detection and 2) reduce the risk of being affected by non-homophily edges. As shown by extensive empirical results and the new results presented in the other replies to reviewers, TAM and its components show promising detection performance on a wide range of datasets, justifying their effectiveness in real-world graph anomaly detection applications.

---

> > ### Comment · Reviewer_uzaW · 2023-08-13
> > **Rebuttal feedback**
> >
> > I have read the authors' rebuttal carefully, and their response addresses my concerns, e.g., one-class homogeneity and obtaining ideal truncated graphs. Based on that, I would like to increase my rating.

---

> > > ### Author Response · Authors · 2023-08-14
> > > **Discussion**
> > >
> > > We're very pleased to hear that our rebuttal helps address your concerns. Thanks a lot for the positive comments and the increased rating!

---

### Official Review · Reviewer_NSnj · 2023-07-09

**Soundness:** 2 fair
**Presentation:** 2 fair
**Contribution:** 2 fair
**Rating:** 4
**Confidence:** 4

**Summary:**

Graph anomaly detection aims to identify abnormal nodes in a given graph. The manuscript argues that the existing graph anomaly detection datasets have one-class homophily where the homophily of normal nodes is much stronger than abnormal nodes. To utilize the one-class homophily phenomenon in graph anomaly detection, a new similarity-based anomaly scoring measure, named local node affinity, is proposed. The proposed method, TAM, learns the parameters of multiple graph neural networks called LAMNets by maximizing the sum of local node affinity on the original graph. The iterative graph truncation technique (NSGT) removes the links between normal and abnormal nodes in each step, and the resulting graphs are fed into LAMNets. During the message passing, LAMNet uses the truncated adjacency matrix. The local node affinity-based anomaly score is measured on the original graph using node representations generated by LAMNets. Experimental results show that TAM achieves high AUROC and AUPRC values compared to seven graph anomaly detection methods on six benchmark datasets.

**Strengths:**

1. NSGT introduces randomness by removing the non-homophily edges in a probabilistic approach. To utilize such randomness, TAM uses an ensemble strategy by performing NSGT multiple times and feeding the resulting graphs to multiple LAMNets. Table 2 in Appendix C shows that the ensemble strategy improves the performance of TAM.

2. The manuscript provides empirical justifications for the major claims. For instance, the homophily distribution in Figure 1(a) supports the one-class homophily phenomenon. Figure 3(a) and Figure 3(b) show that the Euclidean distance between the nodes connected by a non-homophily edge tends to be greater than that between the nodes connected by a homophily edge.

3. Ablation studies with various variants of TAM and hyperparameter sensitivity analyses show that each component of TAM is effective.

**Weaknesses:**

1. It is not guaranteed that the one-class homophily property holds for all graph anomaly detection datasets. For instance, both normal and abnormal nodes in the YelpChi-RUR dataset are homophilic, as shown in Figure 1 (d) in Appendix A.

2. The performance of NSGT might depend on the quality of the node attributes. NSGT performs the graph truncation by considering the node attributes of the original graph. As mentioned in lines 170-171, if the original node features contain many irrelevant attributes, NSGT might not work well. Furthermore, features of abnormal nodes can be intentionally camouflaged [1, 2].

   [1] Liu et al., Alleviating the Inconsistency Problem of Applying Graph Neural Network to Fraud Detection, SIGIR 2020.

   [2] Dou et al., Enhancing Graph Neural Network-based Fraud Detectors against Camouflaged Fraudsters, CIKM 2020.

- One suggestion is to utilize the node representation vectors calculated by LAMNets. In addition, an attribute selection strategy can be introduced to remove irrelevant or redundant node attributes.

3. While this manuscript assumes an unsupervised learning setting, one of my concerns is that most state-of-the-art methods assume a semi-supervised learning setting. Indeed, in many real practices, some supervision can be provided for graph anomaly detection, e.g., labeling undesirable nodes or patterns. Given this, the benefit of the proposed method can be limited.

4. Existing studies [3] convert YelpChi by merging different relations into a single relation. Why YelpChi-RUR and Amazon-UPU are used instead of the full datasets? Instead of selecting one particular relation, the authors can utilize all relations and treat them as a single relation.

   [3] Chen et al., GCCAD: Graph Contrastive Learning for Anomaly Detection, TKDE 2022.

5. The statistics of Amazon and YelpChi differ from the actual ones. For Amazon, the number of nodes is 10,224, and the number of anomalies is 693 (6.78%). For YelpChi, the number of nodes is 23,831, and the number of anomalies is 1,217 (5.11%).

6. Minor Comments:
- Lines 206-207 are confusing. According to lines 206-207, the truncated adjacency matrix instead of the original adjacency matrix is used to optimize LAMNets. However, the local affinity is calculated based on the original adjacency matrix during optimization, as stated in lines 261-262.
- In lines 215-221, it is not specified how to handle the case where $d_{i, max}$ is less than $d_{mean}$.
- In lines 116-117, "such degree" should be modified to "such as degree".

**Questions:**

1. Why does NSGT utilize raw attributes instead of node representations learned from LAMNet?
2. Is there any experimental result on large-scale graph anomaly detection datasets?
3. Isolated nodes can appear after performing NSGT. How does TAM-T compute the anomaly scores of such isolated nodes?

**Limitations:**

1. As the TAM model works solely on the assumption of one-class homophily, it might not perform effectively on the graphs where abnormal nodes are homophilic, e.g., YelpChi-RUR.
2. Since NSGT directly utilizes the raw node features to truncate the graph, the quality of the graph truncation might be affected by the quality of the raw features. However, there can be many irrelevant or redundant attributes in the original features.
3. Due to the definition of local node affinity measure, TAM cannot handle isolated nodes in the graph even though the node features are available.

---

> ### Author Rebuttal · Authors · 2023-08-09
>
> Thank you very much for the constructive comments and questions. We are grateful for the positive comments on our design, empirical justification and ablation study. Please see our detailed one-by-one responses below.
>
> > **Weaknesses/Limitations #1**: Does one-class homophily always hold?
>
> Please refer to our reply to **Global Response to Shared Concern #1** in the overall Author Rebuttal section above for this concern.
>
> > **Weaknesses/Limitations #2, Questions #1**: The dependence of the performance of NSGT/TAM on the quality of raw node attributes.
>
> In TAM, NSGT performs truncation on the graph sequentially based on node attribute similarities to reduce the negative impact of non-homophily edges on message passing. So, it's true that the performance of NSGT (and subsequently TAM) relies on the quality of node attributes, but we show below that performing NSGT directly on the original attributes is a simple, easy-to-use, yet effective way. The two alternative ways you suggested for NSGT do not show clear advantages. We discuss this argument in detail below.
>
> **Using Attribute Selection Before NSGT**. The table below shows the AUC-ROC results of TAM on datasets resulting from the popular unsupervised Laplacian score-based attribute selection (X% means X% top-ranked attributes are selected).
>
> |**Data**|**100%**|**80%**|**60%**|**40%**|**20%**|
> |:----|:----:|:----:|:----:|:----:|:----:|
> |Amazon|0.7064|0.7329|0.7136|0.6924|0.6893|
> |Facebook |0.9144|0.9151|0.8739|0.8802|0.8177|
> |Reddit   |0.6023|0.5845|0.5789|0.5778|0.5664|
> |YelpChi|0.5643|0.5695|0.5749|0.5793|0.5354|
>
> The results show that a careful unsupervised attribute selection can improve the performance of TAM, such as retaining 80% top-ranked attributes in Amazon, Facebook, and YelpChi, but the improvement is often relatively very marginal. Further, if there is any issue in the attribute selection process (e.g., an undesired selection threshold is used), the relevant attributes may also be removed together with the irrelevant ones, leading to degraded performance of TAM, e.g., the case in Reddit. Note that the node attributes in BlogCatalog and ACM are node embedding features, which are mostly all relevant in these two cases. So, we did not include these two datasets in the attribute selection above.
>
> **Using Learned Node Representations from LAMNet**. As suggested by you, we can perform NSGT on the node representations learned by LAMNet, meaning we first run the full TAM on the original graph and then obtain the newly learned node representations from LAMNet as input to re-run TAM. The AUC-ROC results in the table below that this strategy is not as effective as our original TAM that performs NSGT based on raw attributes.
>
> |**Data**|**RAW Attributes**|**LAMNet-based Representations**|
> |:----|:----:|:----:|
> |BlogCatalog|**0.8248**|0.7024|
> |ACM|**0.8878**|0.8670|
> |Amazon|**0.7064**|0.6766|
> |Facebook |0.9144|**0.9164**|
> |Reddit   |**0.6023**|0.5921|
> |YelpChi|**0.5643**|0.5428|
>
> **Handling Camouflage Features**. We evaluate the performance of TAM when there are camouflaged features in the raw attributes. Particularly, we replace 10%/20%/30% randomly sampled original features with camouflages features, in which the feature value of the abnormal nodes is replaced (camouflaged) with the mean feature value of the normal nodes. The AUC-ROC results are shown in the table below, which demonstrate that TAM is robust to a high level of camouflaged features and maintains good superiority over the SOTA models that work on the original features.
>
> |**Data**|**0%**|**10%**|**20%**|**30%**|
> |:----|:----:|:----:|:----:|:----:|
> |BlogCatalog|0.8218|0.8045|0.8022|0.7831|
> |ACM|0.8878|0.8727|0.8688|0.8652|
> |Amazon|0.7064|0.7036|0.6954|0.6838|
> |Facebook |0.9144|0.8870|0.8804|0.8650|
> |Reddit   |0.6023|0.5998|0.5915|0.5876|
> |YelpChi|0.5649|0.5447|0.5271|0.5301|
>
> We will add the above results and discussions into our final paper to provide these important insights for readers.
>
> >**Weaknesses #3**: Unsupervised vs. supervised anomaly detectors?
>
> There are methods from two different paradigms having a different set of pros and cons.
>
> >**Weaknesses #4/Questions #2**: A larger scale of Amazon and YelpChi, and other large datasets?
>
> Please refer to **Global Response to Shared Concern #2** in the overall Author Rebuttal section above.
>
> >**Weaknesses #5, Questions/Limitations #3**: (1) Difference of statistics in YelpChi-RUR and Amazon-UPU? (2) Handling isolated nodes?
>
> (1) We use exactly the same data sources, but we remove primarily isolated nodes in both datasets in data preprocessing, which is the cause of the difference.
>
> (2) There can be two types of isolated nodes. One is the primarily isolated nodes in a graph. Since this kind of nodes does not have any structure information, it has no impact on the graph structure learning, so one commonly used practice is to not consider these nodes in training and evaluation. We also follow this practice. Using only the node attributes to perform anomaly detection will go down to general tabular anomaly detection.
>
> The other one is the isolated nodes that appear after truncating the graph. For this type of isolated nodes, TAM can well handle them. This is because the LAMNet is performed on the truncated graph structure, while the local affinity-based anomaly scoring is done using the original graph structure. Thus, TAM can effectively calculate the anomaly scores for this type of isolated nodes. Normally these nodes would have a small local affinity and a large anomaly score, since all their edges are cut off during our graph truncation. TAM-T has different anomaly scoring from TAM, which is based on the truncated graph to calculate the affinity, so TAM-T takes a simplified approach to deal with isolated nodes: it directly takes the isolated nodes emerging during truncation as anomalies.
>
> **Note that due to space limitation, above we present the AUC-ROC results only. Similar findings can be observed in AUC-PR.**

---

### Author Rebuttal · Authors · 2023-08-09

Dear All Reviewers,

Thank you very much for the time and effort on reviewing our paper, and for the constructive and positive comments. Our rebuttal consists of two part: **Global Response** where we address shared concerns from two or more reviewers and **Individual Response** where we provide a detailed one-to-one response to address your questions/concerns individually.

> **Global Response to Shared Concern #1**: The one-class homophily property may not hold for all graph anomaly detection datasets, such as YelpChi-RUR.

As we show in the paper that the one-class homophily generally holds in popular real-world graph anomaly detection datasets, including datasets with either injected or genuine anomalous nodes. Further, this homophily property also holds for the four large-scale datasets we newly add, T-Finance , Amazon-all, YelpChi-all  and OGB-Protein (see Figure 1(a-d) in the uploaded pdf file for the visualization).

We agree that the one-class homophily may not alway hold, or it is weak in some datasets. YelpChi-RUR is an example of the latter case. As shown in Figure 1 (d) in Appendix A, the homophily distribution of normal and abnormal nodes is similar in YelpChi-RUR, whose pattern is different from the large distribution gap in the other three datasets. Nevertheless, despite having a similar distribution, it is clear that the homophily distribution of normal nodes is still much stronger than that of abnormal nodes on YelpChi-RUR, as shown in the figure. As a result, when applying NSGT for graph truncation, we can still successfully remove the non-homophily edges that connect normal and abnormal nodes with a higher probability than the homophily edges, resulting in a truncated graph with stronger one-class homophily (see Figure 1(e)(f) in the uploaded pdf file for a comparison of homophily before and after the truncation). This enables better detection performance of our method (an AUC of 0.5643) on YelpChi-RUR compared to SOTA models (e.g., an AUC of 0.4956 for the best competing method).

It should be noted that the widely-used general homophily property can also do not hold in some real-world datasets, such as those for node classification [Ref1-Ref3], where many connected nodes are from different classes. This situation can also apply to our proposed one-class homophily property. However, as the first work on the one-class homophily, we focus on the justification of this property and a novel approach to effectively utilize it for popular graph anomaly detection datasets, including those that has a weaker one-class homophily property and datasets of diverse characteristics. This offers a new perspective to design graph anomaly detection methods, promoting the development of more effective graph anomaly detection algorithms that do not use the overwhelming reconstruction/self-supervised frameworks. Exploring the possible presence of heterophily phenomenon in graph anomaly detection datasets and how to effectively avoid such issues in the detection algorithms (including our method TAM) would be some important, interesting follow-up problems to be addressed.

**References**
- [Ref1] Graph neural networks with heterophily.  AAAI.
- [Ref2] Finding global homophily in graph neural networks when meeting heterophily.  PMLR.
- [Ref3] Graph neural networks for graphs with heterophily: A survey. arXiv preprint arXiv:2202.07082.

> **Global Response to Shared Concern #2**:  Any empirical support from results on large-scale datasets?

Inspired by your comments, we add more experiments on two dataset with a large set of edges, Amazon-all and YelpChi-all by treating the different relations as a single relation. In addition, we add another two large datasets T-Finance and OGB-Proteins with a large set of edges and/or nodes. Their key statistics are given in Table A1 below.

```
Table A1. Key Statistic of New Datasets.
```
|**Data**|**Nodes**|**Edges**|**Features**|**Anomaly**|
|:----|:----:|:----:|:----:|:----:|
|Amazon-all| 11,944|4,398,392 |25|9.5%|
|YelpChi-all| 45,954|3,846,979|32|14.5%|
|T-Finance | 39,357|21,222,543|10|4.6%|
|OGB-Protein |132,534 |39,561,252 |8|4.5%|

The results in the two tables below show that TAM also obtains large detection improvement over four best-performing competing methods across four datasets (comparably well to DOMINANT in AUC-PR on OGB-Protein).

```
Table A2. AUC-ROC Results.
```
|**Data**|**Amazon-all**|**YelpChi-all**|**T-Finance**|**OGB-Proteins**|
|:----|:----:|:----:|:----:|:----:|
|DOMINANT|0.6937|0.5390|0.5380|0.7267|
|ComGA|0.7154|0.5352|0.5542|0.7134|
|CoLA|0.2614|0.4801|0.4829|0.7142|
|SL-GAD|0.2728|0.5551|0.4648|0.7371|
|TAM|**0.8476**|**0.5818**|**0.6175**|**0.7449**|
```
Table A3. AUC-PR Results.
```
|**Data**|**Amazon-all**|**YelpChi-all**|**T-Finance**|**OGB-Proteins**|
|:----|:----:|:----:|:----:|:----:|
|DOMINANT|0.1015| 0.1638|0.0474|**0.2217**|
|ComGA|0.1854|0.1658|0.0481|0.1554|
|CoLA| 0.0516|0.1361|0.0410| 0.1349|
|SL-GAD|0.0444|0.1711|0.0386|0.1771|
|TAM|**0.4346**|**0.1886**|**0.0547**|0.2173|

Performing experiments of using TAM and its competing methods on datasets with a even larger number of nodes, such as DGraph that has millions of nodes, requires a GPU server with an extremely large memory. We're currently lack of those computing resource to do that. Note that many existing graph anomaly detection methods, and GNN-based methods for other tasks as well, have similar issues w.r.t. scaleup to such large datasets. We will try to add results on DGraph in the final paper by tapping Amazon cloud computing service for the experiments.

As for **Individual Response**, we have provided a detailed one-by-one response to answer/address your questions/concerns after your individual review.

We very much hope our response has clarified the confusions, and addressed the concerns. We're more than happy to take any further questions if otherwise. Please kindly advise, and have a great week ahead!

Best regards,

Authors of Paper 9585

---

### Comment · Area_Chair_g6rY · 2023-08-18
**Please read the rebuttals and add a comment to acknowledge**

Dear Reviewers,

Many thanks for your time and efforts on this paper. As the end of discussion is coming soon, please read the rebuttal to see if your concerns/questions are properly resolved and add a comment to acknowledge that you have read the rebuttal.


Many Thanks.

---

### Decision · Program_Chairs · 2023-09-21

**Decision:**

Accept (poster)

**Comment:**

I agree with most of the reviewers to recommend the acceptance of this paper after the detailed rebuttal and extra experimental results from the authors to address many questions and issues raised by reviewers. But in the meanwhile, I would encourage the authors to take the reviewers' feedbacks in the preparation of the camera ready. For example, in additional to empirical justification, it will be good to provide a little bit more insight and analysis to explain about why it might be more robust to handle camouflaged features.